# Multi-Actor Multi-Critic Deep Deterministic Reinforcement Learning with a Novel Q-Ensemble Method

## Abstract

Reinforcement learning has gathered much attention in recent years due to its rapid development and rich applications, especially on control systems and robotics. When tackling real-world applications with reinforcement learning method, the corresponded Markov decision process may have huge discrete or even continuous state/action space. Deep reinforcement learning has been studied for handling these issues through deep learning for years, and one promising branch is the actor-critic architecture. Many past studies leveraged multiple critics to enhance the accuracy of evaluation of a policy for addressing the overestimation and underestimation issues. However, few studies have considered the architecture with multiple actors together with multiple critics. This study proposes a novel multi-actor multi-critic (MAMC) deep deterministic reinforcement learning method. The proposed method has three main features, including selection of actors based on non-dominated sorting for exploration with respect to skill and creativity factors, evaluation for actors and critics using a quantile-based ensemble strategy, and exploiting actors with best skill factor. Theoretical analysis proves the learning stability and bounded estimation bias for the MAMC. The present study examines the performance on a well-known reinforcement learning benchmark MuJoCo. Experimental results show that the proposed framework outperforms state-of-the-art deep deterministic based reinforcement learning methods. Experimental analysis also indicates the proposed components are effective. Empirical analysis further investigates the validity of the proposed method, and shows its benefit on complicated problems.

## 1 Introduction

Reinforcement learning (RL) has been studied for decades that is proved powerful when dealing with problems and applications which is assumed or is able to be formulated as a Markov decision process [25]. Numerous applications have been successfully solved by RL methods such as playing board games [26], training large-language model [24], and controlling humanoid [29]. RL methods are of several types, including value-based approach, policy gradient approach, policy optimization approach, and actor-critic approach [28]. This study focus on actor-critic based RL methods due to its nice performance on continuous control problems.

A common issue in RL is the huge or infinite space of states/actions, making conventional tabular methods inapplicable, and a straight forward solution is to construct approximation function for space transformation. As the rapid growth in high performance computing and deep learning [3], leveraging deep learning for building mapping function in RL methods, forming deep reinforcement learning (DRL), becomes possible. One representative method of DRL is the deep Q-learning (DQN)

Table 1: A compilation of some recent proposed actor-critic architectures according to the number of actors and critics

| Method | | #Critics | | |
| | | Single | Double | Multiple |
| --- | --- | --- | --- | --- |
| #Actors | Single | DQN[23], DDPG[19]. | TD3[9], SAC[10], OAC[7]. | REDQ[6], MD3[34], QWPVOP[17]. |
| | Double | - | DARC[21]. | MA-TD3[18]. |
| | Multiple | - | - | SUNRISE[16], MA-DDPG[18]. |

[23], which adopted deep convolution neural network to estimate the state-action function (a.k.a. $Q$ function).

Advanced issues in deep reinforcement learning have been studied and investigated in past years [11]. Essential issues covers learning stability [2, 19, 9], estimation accuracy for handling issues of overestimation [4, 20], underestimation [7, 35] or both [32, 1, 15], sampling efficiency [10, 36, 22**?** ], ensemble learning [21, 6, 16, 14, 34] and so forth, and hybridization of components for addressing these issues is proved to gain effectiveness and learning efficiency [12]. It is worth noting that these issues are highly correlated so that ensemble learning could handle estimation accuracy, which may bring learning stability and sampling efficiency, and thus results in better performance and convergence.

This study proposes a novel method: multiple-actors-multiple-critics (MAMC) deep deterministic reinforcement learning to address the above issues. The main features of the MAMC are threefold: 1) The MAMC manipulates multiple actors and critics in a current manner without predetermined relations, 2) The MAMC evaluates actors and critics as per a quantile-based ensemble strategy, and 3) The MAMC selects actors for exploration in learning on the basis of non-dominated sorting with respect to skill and creativity factors. The emerging MAMC is capable of facilitating nice exploration among multiple actors in the meantime improving and smoothing the learning of critics, which is key to stabilize the guiding force to actors.

The main contributions are listed as follows:

- Devise a parametric quantile-based ensemble estimator considering multiple actors and multiple critics for the target values of critics learning
- Design an actor evaluation and selection approach based on skill and creativity factors for exploration and exploitation
- Theoretically prove the MAMC has stable learning and bounded estimation bias
- Empirically examine the quality and validity of the MAMC, and investigate the run-time behavior of MAMC by inspecting into the proposed components

The rests of this study are organized as follows. Section 2 reviews recent RL methods under actor-critic architectures, and Section 3 introduces preliminaries of this study. Sections 4 and 5 in turn gives details and theoretical analysis for the proposed method. Section 6 examines the effectiveness for the proposed method. Section 7 draws conclusions.

## 2 Related Work

The actor-critic architecture is proposed by Konda and Tsitsiklis [13]. Table 1 compiles six out nine categories of actor-critic architectures in terms of the number of actors and critics for some recent proposed actor-critic-based RL methods. To the best of our knowledge, it is merely no study for actor-critic architectures with fewer number of actors than of critics.

**SASC.** For single-actor single-critic (SASC) architecture, a representative study is the deep Q-network (DQN) [23], which introduced deep convolution neural network into RL and brought nice performance on Atari. Another famous SASC-based method is the deep deterministic policy gradient (DDPG) [19]. DDPG ameliorated the learning stability and efficiency of DQN by combining deep learning with policy gradient for solving control problems with continuous action space.

**SADC.** Beyond SASC, lots of methods are proposed with a single actor and double critics, noted as SADC, for solving the issues of overestimation and exploration. In [9], a twin delayed deep deterministic policy gradient (TD3) was proposed. TD3 improved DDPG by adopting two critic networks, where a minimum of the corresponded two target networks are served as the computational basis of target value. TD3 also proposed the delayed update of actor, i.e., a lower update frequency than critics, for stabilizing the learning of the actor. Soft Actor-Critic (SAC) considered stochastic policy and introduced soft value function for training the two critics of soft Q-function [10]. Specifically, SAC trained a stochastic policy network to transform noise to an action for a given state as condition, and the training depends on a policy gradient for maximizing the randomness of the resulting actions, and the approximated Q values obtained from the minimum of the two critics as TD3. Different from TD3, SAC trained the two critics independently according to the target soft value function network, which is soft-updated by the soft value function network, while the soft value function network is trained by minimizing the different to the target value, which is calculated as the expectation of state-action value of the minimum of the two critics over action given by the actor. Optimistic Actor-Critic (OAC) further pointed out the issues of inefficient exploration owing to insufficient pessimistic in TD3 and SAC, and proposed an amelioration to guide the exploration according to the approximated lower and upper bounds of the state-action value function [7].

**SAMC.** From the observation of improvement from SASC to SADC, many methods considered increasing the number of critics for improving the estimation accuracy, forming the single-actor multi-critic architecture (SAMC). Randomized ensembled double Q-learning (REDQ) [6] estimated the state-action value using the same strategy of minimum the same as TD3, yet the two critics were randomly selected from a pool of critics. REDQ also introduced a high update-to-date (UTD) ratio of 20 to address the issue of sample efficiency. For addressing the estimation accuracy issue, quasi-median Q-learning (QMQ) used the quasi-median among multiple state-action values, each of which from a critic, to estimate the state-action value, and applied on TD3, forming the QMD3. The QMD3 trained actor with delay the same as TD3, but each update is guided by all critics rather than a single one for exploration improvement. Weakly pessimistic value estimation and optimistic policy optimization (WPVOP) [17] proposed weakly pessimistic value estimation and optimistic policy optimization; the former increased and smoothed the lower confidence bound, whilst the latter encourages and increases the state-action value, as the maximum action of minimum state-action values, if the distribution of state-action values with different actions on a given state is centralized, i.e., the standard deviation less than some threshold.

**DADC.** From single actor to double actors, double actors and regularized critics (DARC) [21] adopted double actors as well as double critics (DADC) and proposed soft target value as a linear combination of the minimum and maximum state-action values of the two actions given by two target actors, each of which is a minimum over two target critics. DARC revised the loss function adopted in TD3 by introducing a weighted regularization term of cross-critic error, i.e., the difference between the two critics.

**DAMC.** A multi-actor mechanism (MAM) was proposed in [18]. The MAM was composed of two components, i.e., obsolescence technology (OT) and Q-value weighting technology (WT). For each actor, the OT adaptively adjust the probabilities of interacting with environment according to the accumulated reward in an episode. For each critic, the WT estimated the target of state-action value by a linear combination of state-action values achieved by the corresponding actor, and by the other actors. The MAM was integrated into TD3 (MA-TD3) with double actors and four critics, forming the double-actor multi-critic architecture (DAMC).

**MAMC.** For multi-actor multi-critic (MAMC) architecture, the MAM was also integrated into DDPG (MA-DDPG) with three actors and three critics, which became another MAMC-based method [18]. An early MAMC method is the Simple UNified framework for ReInforcement learning using enSEmbles (SUNRISE) [16]. SUNRISE manipulated multiple SAC agents, each contained a pair of soft Q-function and an actor. SUNRISE integrated weighted Bellman backup, which decreases the influence from high variance transitions, and upper confidence bound (UCB) exploration [5].

# 3 Preliminaries

Given a Markov decision process $(\mathcal{S}, \mathcal{A}, \mathcal{P}, \mathcal{R}, \gamma)$ with state space $\mathcal{S}$, action space $\mathcal{A}$, a state transition probability $\mathcal{P}_{s,s'}^a$, a reward function $\mathcal{R}_{s,a} = \mathbb{E}[R_{t+1}|S_t = s, A_t = a]$, and a discount factor $\gamma$,

Table 2: Notation system

| Symbol | Meaning |
| --- | --- |
| $N_A$, $N_C$, $N_{\mathcal{B}}$ | Number of actors, critics, and mini-batch size |
| $\pi_\phi$ | Actor network with parameter $\phi$ |
| $A$, $\tilde{A}$ | Actors and selected actors |
| $C$, $C'$ | Critics and target critics |
| $Q_\theta$ ($Q_{\theta'}$) | Critic (target) network with parameter $\theta$ ($\theta'$) |
| $\mathcal{R}$, $\mathcal{B}$ | Replay buffer and mini-batch |
| $M$ | Sample multiple reuse |
| $(s, a, r, s')$ | Transition from state $s$ to next state $s'$ by action $a$ with reward $r$ |
| $\gamma$ | Discount factor |
| $\vec{J}_s(A)$, $\vec{J}_c(A)$ | Skill and creativity factors of actors $A$ |
| $\prec$ | Crowded-comparison operator |
| $\mathcal{N}(\mu, \sigma)$ | Gaussian distribution with mean $\mu$ and variance $\sigma^2$ |
| $\tau$ | Soft update ratio |

reinforcement learning aims at learning policy $\pi$ to achieve optimal return from rewards. A famous method is the Q-learning [33], which learns a state-action value function for estimating the reward function $\mathcal{R}_{s,a}$

$$
\begin{aligned}
Q^\pi(s, a) &= \mathbb{E}[R_{t+1}|S_t = s, A_t = a] \\
&= \mathbb{E}[r_{t+1} + \gamma Q^\pi(S_{t+1} = s', A_{t+1} = \pi(s_{t+1}))|S_t = s, A_t = a]
\end{aligned} \tag{1}
$$

The estimation forms a Bellman equation, which can be solved by temporal difference (TD) [27, 30] methods. TD methods approximate the expected return by gradually lowering down the TD error, i.e., the difference of returns between the state-action value $Q(s, a)$ and the TD-target $r_{t+1} + \gamma V(s_{t+1})$, where $V(s_{t+1})$ is the state-value function satisfying $V(s_{t+1}) = Q(s_{t+1}, \pi(s_{t+1}))$.

Establishing approximation function to form a mapping from state space to action space $\pi_\phi : \mathcal{S} \to \mathcal{A}$ and a mapping from state space and action space to a real-value $Q_\theta : \mathcal{S} \times \mathcal{A} \to \mathbb{R}$ by deep neural network forms deep reinforcement learning. According to [9], the update of critic then can be made by minimizing the critic loss function:

$$
J_Q(\theta) = \mathbb{E}_{(s,a,r,s') \sim \mathcal{B}}[(Q_\theta(s, a) - r + \gamma V_\phi(s'; \theta'))^2], \tag{2}
$$

subject to

$$
V_\phi(s'; \theta') = Q_{\theta'}(s', \pi_\phi(s') + \epsilon), \tag{3}
$$

where $\theta'$ is the parameters of critic target with soft update, satisfying $\theta' \leftarrow \tau\theta + (1 - \tau)\theta'$, and $\epsilon$ is the policy noise similar to the technique adopted in SARSA learning [28]. The soft update is for stabilizing the learning of critic network using a fixed target. Then, the update of actor is to minimize the actor loss function:

$$
J_\pi(\phi; \theta) = \mathbb{E}_{(s,a,r,s') \sim \mathcal{B}}[-Q_\theta(s, \phi(s))]. \tag{4}
$$

## 4 MAMC

This study proposes a multi-actor-multi-critic architecture-based RL method: the Multi-Actor Multi-Critic deep deterministic reinforcement learning (MAMC). There are three main features in the proposed MAMC, including the adoption of multiple actors and critics without predefined interaction, the quantile-based ensemble estimation, and the selection of actors as per proposed skill and creativity factors for exploration and exploitation. Table 2 provides the notation system used in this study.

### 4.1 The Overall Procedure

Algorithm 1 gives the main procedure of the proposed MAMC. At initialization, the MAMC generates a set of $N_A$ actor networks $A$ and a set of $N_C$ critic networks $C$ with random parameters, and set the parameters of each target network according to the parameters of its corresponded critic network. The replay buffer $\mathcal{R}$ is also initialized by random actions of a predefined size. During each iteration, there are three main stages: critics learning stage, actors learning state, and exploration stage.

---

**Algorithm 1** Main procedure of MAMC

---

1: Initialize a set of $N_A$ actor networks $A$ with random parameters $\{\phi_i\}_{1 \leq i \leq N_A}$
2: Initialize a set of $N_C$ critic networks $C$ with random parameters $\{\theta_j\}_{1 \leq j \leq N_C}$
3: Initialize a set of $N_C$ target networks $C'$ with critics $\theta'_j \leftarrow \theta_j$ for $1 \leq j \leq N_C$
4: Initialize replay buffer $\mathcal{R}$
5: **while** Not Terminated **do**
6:     ▷ **Critics Learning**
7:     $\{\mathcal{B}_j\}_{1 \leq j \leq N_C} \sim \mathcal{R}$         ▷ Sample a mini-batch from replay buffer $R$ for each critic
8:     **for** $m \leftarrow 1$ **to** $M$ **do**         ▷ Sample multiple reuse
9:         Update $\theta_j$ on $\mathcal{B}_j$ according to Eqs. (6) and (7) for $1 \leq j \leq N_C$
10:         Update $\theta'_j$ by soft update for $1 \leq j \leq N_C$
11:     **end for**
12:     ▷ **Actors Learning**
13:     $\{\mathcal{B}_i\}_{1 \leq i \leq N_A} \sim \mathcal{R}$         ▷ Sample a mini-batch from replay buffer $R$ for each actor
14:     **for** $m \leftarrow 1$ **to** $M$ **do**         ▷ Sample multiple reuse
15:         $j \leftarrow m \mod N_C$         ▷ Guided by each critic in turn
16:         Update $\phi_i$ by $\theta_j$ on $\mathcal{B}_i$ according to Eq. (4) for all $1 \leq i \leq N_A$
17:     **end for**
18:     ▷ **Exploration**
19:     $\tilde{A} \leftarrow$ Selection $(\vec{J}_s(A; C), \vec{J}_c(A; C), \prec)$         ▷ Crowded-comparison operator
20:     $(r, s') \leftarrow Env\left(s, a = \pi_{\phi \sim \tilde{A}}(s) + \epsilon\right), \epsilon \sim \mathcal{N}(0, \sigma)$         ▷ Interact with environment
21:     $\mathcal{R} \leftarrow \mathcal{R} \cup (s, a, r, s')$
22:     $\pi^* \leftarrow \arg\min_\phi J_s(\phi; C)$
23: **end while**
24: **return** $\pi^*$

---

### 4.2 Quantile-based Ensemble Estimation

In critics learning stage, $N_C$ sets of mini-batch $\{\mathcal{B}_j\}_{1 \leq j \leq N_C}$ are sampled from the replay buffer $\mathcal{R}$, and each critic is trained on a specific mini-batch for $M$ times for improving the stability.

**Definition 1.** For each transitions $(s, a, r, s') \in \mathcal{B}_j$, the TD-target for $j$th critic $Q_{\theta_j}$ is defined as the median action of the $q$th-quantile among the critic targets:

$$y(s, a) = r + \gamma \hat{V}_A(s'), \tag{5}$$

subject to

$$\begin{aligned}
\hat{V}_A(s'; C') &= \text{Med}(\{\hat{V}_{\phi_i}(s'; C')\}_{1 \leq i \leq N_A}) \\
\hat{V}_{\phi_i}(s'; C') &= \text{Quantile}_q(\{Q_{\theta'_j}(s', \pi_{\phi_i}(s') + \epsilon)\}_{1 \leq j \leq N_C})
\end{aligned}. \tag{6}$$

The critic loss function is therefore defined as

$$J_Q(\theta_j; C') = \mathbb{E}_{(s,a,r,s') \sim \mathcal{B}}[(Q_{\theta_j}(s, a) - r + \gamma V_A(s'; C'))^2]. \tag{7}$$

All the target critics are soft-updated with parameter $\tau$ after one out of $M$ iterations of training, which is capable of sharing information to each target critic from all the other critic targets and bring to the next iteration.

For the learning of actors, the MAMC also sampled $N_A$ sets of mini-batch $\{\mathcal{B}_i\}_{1 \leq i \leq N_A}$ from the replay buffer $\mathcal{R}$ as it does in critics learning stage. The training of each actor $\pi_{\phi_i}$ is in turn guided by each critic $Q_{\theta_j}$ with objective $J_\pi(\phi_i; \theta_j)$ (cf. Eq. (4)) on its mini-batch $\mathcal{B}_i$. The idea of updating $M$ times within a mini-batch for each actor and critics is similar to sample multiple reuse (SMR) proposed in [22], which is able to stabilize the learning sequence.

### 4.3 Actor Evaluation, Exploration, and Exploitation

After training of actors and critics, the exploration stage is to select appropriate actors for interacting with the environment. The evaluation of an actor $\pi_{\phi_i}$ is based on two factors, i.e., skill and creativity, both are determined by the ensemble estimation of state value function.

**Definition 2.** Ensemble estimation of state value function is defined as the $q$th-quantile of state-action value function over critics $C$:

$$\hat{V}_{\phi_i}(s^{(k)}; C) = \text{Quantile}_q(\{Q_{\theta_j}(s^{(k)}, \pi_{\phi_i}(s^{(k)}))\}_{1 \leq j \leq N_C}), \tag{8}$$

where $s^{(k)}$ is the $k$th transition in a mini-batch. The consideration of skill factor guarantees the quality of interaction, whilst the consideration of creativity factor preserves the diversity of interaction.

The skill factor evaluates the optimality of an actor through the scoring ability on the ensemble estimation

$$J_s(\phi_i; C) = N_{\mathcal{B}}^{-1} \sum_{k=1}^{N_{\mathcal{B}}} \hat{V}_{\phi_i}(s^{(k)}), \tag{9}$$

while the creativity factor examines the diversity of an actor on the critics through the closeness of each critic to the ensemble estimation with respect to mean absolute error

$$J_c(\phi_i; C) = N_{\mathcal{B}}^{-1} N_C^{-1} \sum_{k=1}^{N_{\mathcal{B}}} \sum_{j=1}^{N_C} |Q_{\theta_j}(s^{(k)}, \pi_{\phi_i}(s^{(k)})) - \hat{V}_{\phi_i}(s^{(k)})|. \tag{10}$$

Both factors are expectation over a mini-batch. Note that the two factors depends on all the critics as rather than critic targets since the actors are guided by critics. The selection of actors on the two factor hinges upon the crowed-comparison operator [8] by considering the two factors as two objective values. The top-$\sqrt{N_A}$ actors $\tilde{A}$ are selected, which serves as the candidate actors for interaction with the environment. Specifically, an actor is randomly picked from the candidate actors $\tilde{A}$ for determining a single step of interaction with the environment. The MAMC also records an optimal policy with highest skill factor for exploitation at each iteration; that is, the MAMC only returns a single actor for inference due to the efficiency in terms of time and space complexity.

## 5 Theoretical Analysis

This section gives some nice properties for the MAMC. First, the target values obtained by multiple actors are more stable in terms of variance than using a single actor.

**Theorem 1.** *The variance of target values obtained by multiple actors are less than that using a single actor.*

$$\mathbb{V}[\hat{V}_A(s'; C')] \leq \mathbb{V}[\hat{V}_\phi(s'; C')] \tag{11}$$

Similarly, the target values obtained by multiple critics are more stable than using a single critic.

**Theorem 2.** *The variance of target values obtained by multiple critics are less than using a single critic.*

$$\mathbb{V}[\hat{V}_\phi(s'; C')] \leq \mathbb{V}[\hat{V}_\phi(s'; \theta')] \tag{12}$$

Thus, the learning stability of the MAMC, with lowest variance, is greater than SAMC and SASC.

Further, this study investigate the property of estimation error, which is a good metric for indicating the estimation accuracy [21].

**Definition 3.** The estimation error of MAMC is defined as the difference between expectation of estimate values and the expectation of optimal policy $\pi$.

$$\mathcal{E}_{A,C} = \mathbb{E}[\hat{V}_A(s'; C)] - \mathbb{E}[\hat{V}_\phi(s'; C)] \tag{13}$$

Then the MAMC holds the following properties.

**Theorem 3.** *The estimation error of MAMC is between the estimation error of multiple actors with minimum and maximum critics.*

$$\mathcal{E}_{A,Q_{\theta_{\min}}} \leq \mathcal{E}_{A,C} \leq \mathcal{E}_{A,Q_{\theta_{\max}}} \tag{14}$$

**Theorem 4.** *The estimation error of MAMC is between the estimation error of multiple critics with minimum and maximum actors.*

$$\mathcal{E}_{\pi_{\phi_{\min}},C} \leq \mathcal{E}_{A,C} \leq \mathcal{E}_{\pi_{\phi_{\max}},C} \tag{15}$$

Hence, the estimation error of MAMC is in between the maximum and minimum of SAMC and MASC. The proofs of above theorems will be given in supplementary material Section B due to space limitation.

Table 3: Wilcoxon signed rank test for TD3 and DARC compared with the MAMC at early (100k), middle (200k), and late stage (300k). The win/tie/lose denotes the number of environments that the MAMC is significantly superior, equal, and inferior to the corresponding test method.

| Stage (win/tie/lose) | TD3-SMR | DARC-SMR | SAC-SMR | REDQ-SMR |
|---|---|---|---|---|
| 100k | 3/2/0 | 2/2/1 | 3/2/0 | 0/4/1 |
| 200k | 2/3/0 | 1/4/0 | 2/2/1 | 0/4/1 |
| 300k | 2/3/0 | 1/3/1 | 1/4/0 | 0/3/2 |

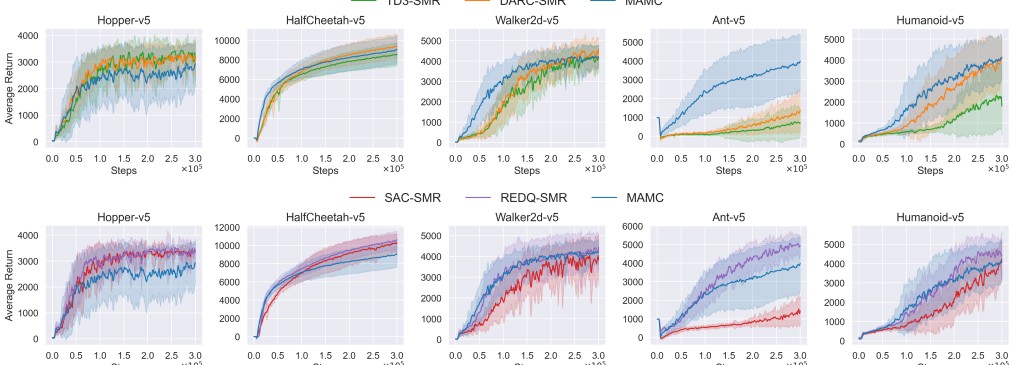

Figure 1: Average return against environment steps for TD3-based and SAC-based methods by comparison with the MAMC on the five environments

## 6  Experimental Results

This section examines the performance of the proposed MAMC method in terms of effectiveness and efficiency through experiments. Further analysis is made for showing the effectiveness of the proposed components, the sensitivity of introduced hyperparameters, and the validity of the MAMC.

### 6.1  Experimental Settings

The experiments are conducted on a set of five test environments chose from the well-known MuJuCo benchmark [31], including Hopper-v5, HalfCheetah-v5, Walker2d-v5, Ant-v5, and Humanoid-v5. These environments are all have continuous state and action spaces with different dimensions. Regarding the dimensionality, the difficulty of each environment can be regarded as either simple (Hopper-v5), medium (HalfCheetah-v5, Walker2d-v5), or hard (Ant-v5, Humanoid-v5). The properties of these environment can be found in the supplementary material Section A.3.

This study selects four state-of-the-art RL methods for performance comparison, including two with deterministic policy: TD3[9] and DARC[21], and two with stochastic policy: SAC[10], REDQ[6]. Both TD3 and SAC used a single actor with two critics. REDQ also adopts a single actor but with ten critics, while DARC exploits two actors and two critics. The proposed MAMC utilizes ten actors and ten critics. An analysis on the number of actors and critics can be found in the supplementary material Section C.3. In addition, as the MAMC considers sample multiple reuse (SMR) [22], all the four test methods are implemented as SMR versions, which are reported with better performance than the original versions, for a fair comparison.

The hyperparameter settings for the four baseline methods follow their original suggestions. The termination criterion is set to 300k environmental steps. All experiments conducted 10 trials, and each trial is an average over twenty seeds for return if not stated. All figures are uniformly smoothed. For significance analysis, this study adopts the Wilcoxon ranksum test with .05 significant level. The error bars are within the range $[\mu - \sigma, \mu + \sigma]$, which are generated by standard deviations with the assumption of normally distributed errors. For more details about the experimental settings, please refer to the supplementary material Section A.

Table 4: Average and standard deviation of return for the MAMC with single-objective and multi-objective actor selection strategies on Ant-v5 over eight trials at early (100k), middle (200k), and late stage (300k). The bold symbol implies the highest value.

| Stage | 100k | | 200k | | 300k | |
|---|---|---|---|---|---|---|
| Ant-v5 | SO | MO | SO | MO | SO | MO |
| Mean | 1980 | **2701** | 2805 | **3611** | 3395 | **4276** |
| Std. | 800 | 1235 | 1014 | 1631 | 1278 | 1260 |

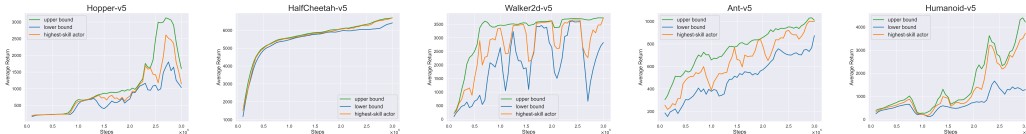

Figure 2: Average return for the best, worst, and skilled (selected) actors in the MAMC in a specific trial on the five test environments

## 6.2 Effectiveness

Table 3 compares the Wilcoxon signed rank test for TD3 and DARC compared with the MAMC at early (100k), middle (200k), and late stage (300k). The details are provided in supplementary material Section C.1. At early stage, the MAMC achieves better quality than the two deterministic methods TD3-SMR and DARC-SMR. Comparing to the two SAC-based methods, the MAMC outperforms SAC-SMR but performs slightly worse than REDQ-SMR on the Hopper-v5 environment. At middle stage, the MAMC still betters TD3-SMR and DARC-SMR, yet the improvement becomes smaller than that at early stage. As for the two SAC-based methods, the trend on REDQ-SMR keeps, while the improvement on SAC-SMR also decreases. At late stage, the lead to TD3-SMR, DARC-SMR, and SAC-SMR further shrinks that the MAMC is slightly superior to TD3-SMR and SAC-SMR, but is comparable to DARC-SMR. The REDQ-SMR further surpasses the MAMC on Humanoid-v5 environment. These results reflect the merits of MAMC at early and middle stage, and the demerit at late stage.

## 6.3 Efficiency

Figure 1 draws the average return against environment steps for TD3-based and SAC-based methods by comparison with the MAMC on the five environments. Compared with TD3-based methods, the MAMC gains faster convergence on the three more complicated environments, i.e., Walker2d-v5, Ant-v5, and Humanoid-v5. Similarly, the MAMC converges faster than SAC-SMR on these three environments, yet the REDQ-SMR converges nicer than the MAMC on all except Walker2d-v5. These results validate the efficiency of the MAMC against the two deterministic method TD3-SMR and DARC-SMR, and the simpler stochastic method SAC-SMR.

## 6.4 Components Analysis

Table 4 lists the average and standard deviation of return for the MAMC with single-objective (MAMC-SO) and multi-objective actor selection strategies on Ant-v5 over eight trials. The MAMC-SO averages the skill and creativity factors and selects the top actors by sorting for exploration. The exploitation selection mechanism for MAMC-SO and MAMC is the same. From the table, the MAMC performs better than MAMC-SO at all the three stages, which verifies the effectiveness of the proposed multi-objective actor selection mechanism.

Figure 2 plots the average return for the best (upper bound), worst (lower bound), and skilled (selected) actors in the MAMC in a specific trial on the five test environments. On HalfCheetah-v5 and Humanoid-v5, the MAMC is capable of selecting good actor approaching the upper bound, to wit, the best actor. For Hopper-v5, Walker2d-v5, and Ant-v5, the MAMC tracks the moving upper bound, and in most of the time the selected actor having quality beyond the average of upper and

Table 5: Average and standard deviation of return for the MAMC with different quantile parameter $q$

| $q$ | $= 0.1$ | $= 0.2$ | $= 0.3$ | $= 0.4$ | $= 0.5$ |
|---|---|---|---|---|---|
| HalfCheetah-v5 | 9119±1077 | 8153±1115 | 9117±1070 | 9191±1043 | **9466**±1256 |
| Walker2d-v5 | 3188±1516 | **4324**±1038 | 4083±927 | 3385±1039 | 1406±561 |

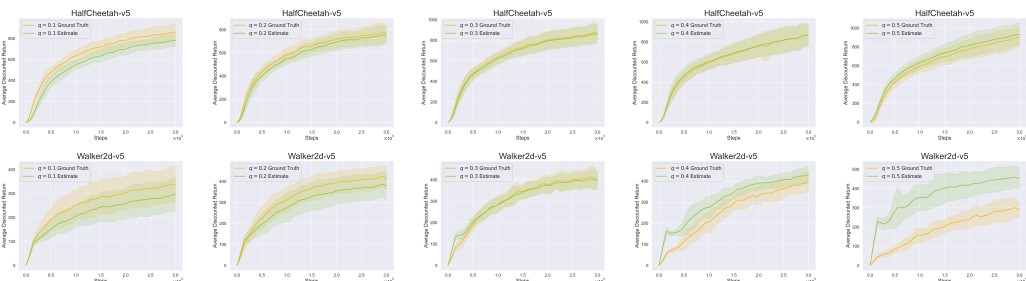

Figure 3: Estimated and ground-truth average discounted return against environment steps for the MAMC with different quantile parameters $q$ on HalfCheetah-v5 and Walker2d-v5

lower bounds. These results validate the effectiveness of the proposed skill factor for actor selection for exploitation.

## 6.5 Sensitivity Analysis

Figure 3 plots the estimated and ground-truth average discounted return against environment steps for MAMC with different quantile parameters $q$ on HalfCheetah-v5 and Walker2d-v5. It is obvious that the estimated value increases as $q$ increases; the best $q$ in terms of the smallest distance to ground-truth value is 0.4 for HalfCheetah-v5 and 0.3 for Walker2d-v5. However, the values are inconsistent to the best $q$ in terms of the average return, which is 0.5 for HalfCheetah-v5 and 0.2 for Walker2d-v5 (cf. Table 5). That is, the setting of quantile parameters $q$ should also considers the environmental preferences of optimism and pessimism. In general, a range between 0.2 and 0.3 is a good setting for environments which favor pessimism, and a value between 0.3 and 0.4 is nice for optimism cases; thus, a robust $q$ value may near 0.3, but the best one for a specific environment still needs to be investigated.

## 6.6 Validity Analysis

The proposed MAMC is based on deterministic policy, and the results have shown that the MAMC can ameliorate the performance of TD3-SMR and REDQ-SMR. The MAMC is also beneficial in comparison to SAC-SMR, a simple but powerful method with stochastic policy. Past studies have discovered the potential of stochastic policy over deterministic policy, and this may be the weakness of the MAMC, which is considered as the main reason to be surpassed by REDQ-SMR.

## 7 Conclusions

This study proposes a multi-actor multi-critic deep deterministic reinforcement learning method. The MAMC includes a selection of actors for exploration using skill and creativity factors, an ensemble target value based on a predefined quantile parameter, and a selection of best actor regarding skill factor for exploitation. Theoretical analysis proves the MAMC having bounded estimation error, and learning stability over SAMC and MASC. From experimental results, the MAMC excels TD3-SMR, DARC-SMR, and SAC-SMR with better quality and faster convergence on the selected environments in MuJoCo. The validity analysis shows a weakness of deterministic based method and is also a possible future extension. Another promising orientation for future research is to adapt the quantile parameter to address the issue of estimation accuracy by balancing optimism and pessimism.

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

# NeurIPS Paper Checklist

1. **Claims**

   Question: Do the main claims made in the abstract and introduction accurately reflect the paper's contributions and scope?

   Answer: [Yes]

   Justification: The abstract and introduction clearly state the claims made, including the contributions made in the paper and important assumptions and limitations.

   Guidelines:

   - The answer NA means that the abstract and introduction do not include the claims made in the paper.
   - The abstract and/or introduction should clearly state the claims made, including the contributions made in the paper and important assumptions and limitations. A No or NA answer to this question will not be perceived well by the reviewers.
   - The claims made should match theoretical and experimental results, and reflect how much the results can be expected to generalize to other settings.
   - It is fine to include aspirational goals as motivation as long as it is clear that these goals are not attained by the paper.

2. **Limitations**

   Question: Does the paper discuss the limitations of the work performed by the authors?

   Answer: [Yes]

   Justification: The limitations of the proposed work have been discussed in Section 6.6.

   Guidelines:

   - The answer NA means that the paper has no limitation while the answer No means that the paper has limitations, but those are not discussed in the paper.
   - The authors are encouraged to create a separate "Limitations" section in their paper.
   - The paper should point out any strong assumptions and how robust the results are to violations of these assumptions (e.g., independence assumptions, noiseless settings, model well-specification, asymptotic approximations only holding locally). The authors should reflect on how these assumptions might be violated in practice and what the implications would be.
   - The authors should reflect on the scope of the claims made, e.g., if the approach was only tested on a few datasets or with a few runs. In general, empirical results often depend on implicit assumptions, which should be articulated.
   - The authors should reflect on the factors that influence the performance of the approach. For example, a facial recognition algorithm may perform poorly when image resolution is low or images are taken in low lighting. Or a speech-to-text system might not be used reliably to provide closed captions for online lectures because it fails to handle technical jargon.
   - The authors should discuss the computational efficiency of the proposed algorithms and how they scale with dataset size.
   - If applicable, the authors should discuss possible limitations of their approach to address problems of privacy and fairness.
   - While the authors might fear that complete honesty about limitations might be used by reviewers as grounds for rejection, a worse outcome might be that reviewers discover limitations that aren't acknowledged in the paper. The authors should use their best judgment and recognize that individual actions in favor of transparency play an important role in developing norms that preserve the integrity of the community. Reviewers will be specifically instructed to not penalize honesty concerning limitations.

3. **Theory assumptions and proofs**

   Question: For each theoretical result, does the paper provide the full set of assumptions and a complete (and correct) proof?

   Answer: [Yes]

Justification: The assumptions and proofs are given in supplementary material Section B.

Guidelines:

- The answer NA means that the paper does not include theoretical results.
- All the theorems, formulas, and proofs in the paper should be numbered and cross-referenced.
- All assumptions should be clearly stated or referenced in the statement of any theorems.
- The proofs can either appear in the main paper or the supplemental material, but if they appear in the supplemental material, the authors are encouraged to provide a short proof sketch to provide intuition.
- Inversely, any informal proof provided in the core of the paper should be complemented by formal proofs provided in appendix or supplemental material.
- Theorems and Lemmas that the proof relies upon should be properly referenced.

4. **Experimental result reproducibility**

Question: Does the paper fully disclose all the information needed to reproduce the main experimental results of the paper to the extent that it affects the main claims and/or conclusions of the paper (regardless of whether the code and data are provided or not)?

Answer: [Yes]

Justification: All the information needed to reproduce the main experimental results of the paper is given in Section 6.1 and supplementary material Section A. The code used in this paper along with instructions containing the exact command and environment needed to run to reproduce the results are also provided in anonymized github page at



`https://github.com/NobodyAcademic/MAMC`



Guidelines:

- The answer NA means that the paper does not include experiments.
- If the paper includes experiments, a No answer to this question will not be perceived well by the reviewers: Making the paper reproducible is important, regardless of whether the code and data are provided or not.
- If the contribution is a dataset and/or model, the authors should describe the steps taken to make their results reproducible or verifiable.
- Depending on the contribution, reproducibility can be accomplished in various ways. For example, if the contribution is a novel architecture, describing the architecture fully might suffice, or if the contribution is a specific model and empirical evaluation, it may be necessary to either make it possible for others to replicate the model with the same dataset, or provide access to the model. In general. releasing code and data is often one good way to accomplish this, but reproducibility can also be provided via detailed instructions for how to replicate the results, access to a hosted model (e.g., in the case of a large language model), releasing of a model checkpoint, or other means that are appropriate to the research performed.
- While NeurIPS does not require releasing code, the conference does require all submissions to provide some reasonable avenue for reproducibility, which may depend on the nature of the contribution. For example
  (a) If the contribution is primarily a new algorithm, the paper should make it clear how to reproduce that algorithm.
  (b) If the contribution is primarily a new model architecture, the paper should describe the architecture clearly and fully.
  (c) If the contribution is a new model (e.g., a large language model), then there should either be a way to access this model for reproducing the results or a way to reproduce the model (e.g., with an open-source dataset or instructions for how to construct the dataset).
  (d) We recognize that reproducibility may be tricky in some cases, in which case authors are welcome to describe the particular way they provide for reproducibility. In the case of closed-source models, it may be that access to the model is limited in some way (e.g., to registered users), but it should be possible for other researchers to have some path to reproducing or verifying the results.

5. **Open access to data and code**

   Question: Does the paper provide open access to the data and code, with sufficient instructions to faithfully reproduce the main experimental results, as described in supplemental material?

   Answer: [Yes]

   Justification: The code used in this paper along with instructions containing the exact command and environment needed to run to reproduce the results are also provided in anonymized github page at

   $$\texttt{https://github.com/NobodyAcademic/MAMC}$$

   Guidelines:

   - The answer NA means that paper does not include experiments requiring code.
   - Please see the NeurIPS code and data submission guidelines (`https://nips.cc/public/guides/CodeSubmissionPolicy`) for more details.
   - While we encourage the release of code and data, we understand that this might not be possible, so "No" is an acceptable answer. Papers cannot be rejected simply for not including code, unless this is central to the contribution (e.g., for a new open-source benchmark).
   - The instructions should contain the exact command and environment needed to run to reproduce the results. See the NeurIPS code and data submission guidelines (`https://nips.cc/public/guides/CodeSubmissionPolicy`) for more details.
   - The authors should provide instructions on data access and preparation, including how to access the raw data, preprocessed data, intermediate data, and generated data, etc.
   - The authors should provide scripts to reproduce all experimental results for the new proposed method and baselines. If only a subset of experiments are reproducible, they should state which ones are omitted from the script and why.
   - At submission time, to preserve anonymity, the authors should release anonymized versions (if applicable).
   - Providing as much information as possible in supplemental material (appended to the paper) is recommended, but including URLs to data and code is permitted.

6. **Experimental setting/details**

   Question: Does the paper specify all the training and test details (e.g., data splits, hyperparameters, how they were chosen, type of optimizer, etc.) necessary to understand the results?

   Answer: [Yes]

   Justification: The experimental setting are presented in Section 6.1. The full details are provided in the supplementary material Section A.

   Guidelines:

   - The answer NA means that the paper does not include experiments.
   - The experimental setting should be presented in the core of the paper to a level of detail that is necessary to appreciate the results and make sense of them.
   - The full details can be provided either with the code, in appendix, or as supplemental material.

7. **Experiment statistical significance**

   Question: Does the paper report error bars suitably and correctly defined or other appropriate information about the statistical significance of the experiments?

   Answer: [Yes]

   Justification: The error bars are within the range $[\mu - \sigma, \mu + \sigma]$, which are generated by standard deviations with the assumption of normally distributed errors. The statistical significance of the experiments is examined by the Wilcoxon ranksum test with .05 significance level. These are mentioned in Section 6.1.

   Guidelines:

- The answer NA means that the paper does not include experiments.
- The authors should answer "Yes" if the results are accompanied by error bars, confidence intervals, or statistical significance tests, at least for the experiments that support the main claims of the paper.
- The factors of variability that the error bars are capturing should be clearly stated (for example, train/test split, initialization, random drawing of some parameter, or overall run with given experimental conditions).
- The method for calculating the error bars should be explained (closed form formula, call to a library function, bootstrap, etc.)
- The assumptions made should be given (e.g., Normally distributed errors).
- It should be clear whether the error bar is the standard deviation or the standard error of the mean.
- It is OK to report 1-sigma error bars, but one should state it. The authors should preferably report a 2-sigma error bar than state that they have a 96% CI, if the hypothesis of Normality of errors is not verified.
- For asymmetric distributions, the authors should be careful not to show in tables or figures symmetric error bars that would yield results that are out of range (e.g. negative error rates).
- If error bars are reported in tables or plots, The authors should explain in the text how they were calculated and reference the corresponding figures or tables in the text.

8. **Experiments compute resources**

Question: For each experiment, does the paper provide sufficient information on the computer resources (type of compute workers, memory, time of execution) needed to reproduce the experiments?

Answer: [Yes]

Justification: All the experiments are conducted on a server with Intel Xeon W7-2475X CPU (with 2.6 GHz clock rate, 20 cores and 40 hyperthreads), two NVIDIA RTX 4090 GPU cards (each with 24GB memory), and 128 GB main memory. The description is revealed in the supplementary material Section A.2.

Guidelines:

- The answer NA means that the paper does not include experiments.
- The paper should indicate the type of compute workers CPU or GPU, internal cluster, or cloud provider, including relevant memory and storage.
- The paper should provide the amount of compute required for each of the individual experimental runs as well as estimate the total compute.
- The paper should disclose whether the full research project required more compute than the experiments reported in the paper (e.g., preliminary or failed experiments that didn't make it into the paper).

9. **Code of ethics**

Question: Does the research conducted in the paper conform, in every respect, with the NeurIPS Code of Ethics `https://neurips.cc/public/EthicsGuidelines`?

Answer: [Yes]

Justification: The authors have checked the NeurIPS Code of Ethics and make sure the conducted research in this paper conform it in every respect.

Guidelines:

- The answer NA means that the authors have not reviewed the NeurIPS Code of Ethics.
- If the authors answer No, they should explain the special circumstances that require a deviation from the Code of Ethics.
- The authors should make sure to preserve anonymity (e.g., if there is a special consideration due to laws or regulations in their jurisdiction).

10. **Broader impacts**

Question: Does the paper discuss both potential positive societal impacts and negative societal impacts of the work performed?

Answer: [No]

Justification: This paper is a foundational research and not tied to particular applications.

Guidelines:

- The answer NA means that there is no societal impact of the work performed.
- If the authors answer NA or No, they should explain why their work has no societal impact or why the paper does not address societal impact.
- Examples of negative societal impacts include potential malicious or unintended uses (e.g., disinformation, generating fake profiles, surveillance), fairness considerations (e.g., deployment of technologies that could make decisions that unfairly impact specific groups), privacy considerations, and security considerations.
- The conference expects that many papers will be foundational research and not tied to particular applications, let alone deployments. However, if there is a direct path to any negative applications, the authors should point it out. For example, it is legitimate to point out that an improvement in the quality of generative models could be used to generate deepfakes for disinformation. On the other hand, it is not needed to point out that a generic algorithm for optimizing neural networks could enable people to train models that generate Deepfakes faster.
- The authors should consider possible harms that could arise when the technology is being used as intended and functioning correctly, harms that could arise when the technology is being used as intended but gives incorrect results, and harms following from (intentional or unintentional) misuse of the technology.
- If there are negative societal impacts, the authors could also discuss possible mitigation strategies (e.g., gated release of models, providing defenses in addition to attacks, mechanisms for monitoring misuse, mechanisms to monitor how a system learns from feedback over time, improving the efficiency and accessibility of ML).

11. **Safeguards**

Question: Does the paper describe safeguards that have been put in place for responsible release of data or models that have a high risk for misuse (e.g., pretrained language models, image generators, or scraped datasets)?

Answer: [No]

Justification: This paper provides open access to the source code under pyTorch and NumPy Licenses without warranty of any kind.

Guidelines:

- The answer NA means that the paper poses no such risks.
- Released models that have a high risk for misuse or dual-use should be released with necessary safeguards to allow for controlled use of the model, for example by requiring that users adhere to usage guidelines or restrictions to access the model or implementing safety filters.
- Datasets that have been scraped from the Internet could pose safety risks. The authors should describe how they avoided releasing unsafe images.
- We recognize that providing effective safeguards is challenging, and many papers do not require this, but we encourage authors to take this into account and make a best faith effort.

12. **Licenses for existing assets**

Question: Are the creators or original owners of assets (e.g., code, data, models), used in the paper, properly credited and are the license and terms of use explicitly mentioned and properly respected?

Answer: [Yes]

Justification: All the code packages and datasets used in this paper are cited and/or credited properly.

Guidelines:

- The answer NA means that the paper does not use existing assets.
- The authors should cite the original paper that produced the code package or dataset.
- The authors should state which version of the asset is used and, if possible, include a URL.
- The name of the license (e.g., CC-BY 4.0) should be included for each asset.
- For scraped data from a particular source (e.g., website), the copyright and terms of service of that source should be provided.
- If assets are released, the license, copyright information, and terms of use in the package should be provided. For popular datasets, `paperswithcode.com/datasets` has curated licenses for some datasets. Their licensing guide can help determine the license of a dataset.
- For existing datasets that are re-packaged, both the original license and the license of the derived asset (if it has changed) should be provided.
- If this information is not available online, the authors are encouraged to reach out to the asset's creators.

13. **New assets**

Question: Are new assets introduced in the paper well documented and is the documentation provided alongside the assets?

Answer: [Yes]

Justification: This paper provides open access to the source code under pyTorch and NumPy Licenses, which is mentioned in an anonymized github page at

$$\text{https://github.com/...}$$

Guidelines:

- The answer NA means that the paper does not release new assets.
- Researchers should communicate the details of the dataset/code/model as part of their submissions via structured templates. This includes details about training, license, limitations, etc.
- The paper should discuss whether and how consent was obtained from people whose asset is used.
- At submission time, remember to anonymize your assets (if applicable). You can either create an anonymized URL or include an anonymized zip file.

14. **Crowdsourcing and research with human subjects**

Question: For crowdsourcing experiments and research with human subjects, does the paper include the full text of instructions given to participants and screenshots, if applicable, as well as details about compensation (if any)?

Answer: [NA]

Justification: The paper does not involve crowdsourcing nor research with human subjects.

Guidelines:

- The answer NA means that the paper does not involve crowdsourcing nor research with human subjects.
- Including this information in the supplemental material is fine, but if the main contribution of the paper involves human subjects, then as much detail as possible should be included in the main paper.
- According to the NeurIPS Code of Ethics, workers involved in data collection, curation, or other labor should be paid at least the minimum wage in the country of the data collector.

15. **Institutional review board (IRB) approvals or equivalent for research with human subjects**

Question: Does the paper describe potential risks incurred by study participants, whether such risks were disclosed to the subjects, and whether Institutional Review Board (IRB) approvals (or an equivalent approval/review based on the requirements of your country or institution) were obtained?

Answer: [NA]

Justification: The paper does not involve crowdsourcing nor research with human subjects.

Guidelines:

- The answer NA means that the paper does not involve crowdsourcing nor research with human subjects.
- Depending on the country in which research is conducted, IRB approval (or equivalent) may be required for any human subjects research. If you obtained IRB approval, you should clearly state this in the paper.
- We recognize that the procedures for this may vary significantly between institutions and locations, and we expect authors to adhere to the NeurIPS Code of Ethics and the guidelines for their institution.
- For initial submissions, do not include any information that would break anonymity (if applicable), such as the institution conducting the review.

16. **Declaration of LLM usage**

Question: Does the paper describe the usage of LLMs if it is an important, original, or non-standard component of the core methods in this research? Note that if the LLM is used only for writing, editing, or formatting purposes and does not impact the core methodology, scientific rigorousness, or originality of the research, declaration is not required.

Answer: [NA]

Justification: The core method development in this research does not involve LLMs as any important, original, or non-standard components.

Guidelines:

- The answer NA means that the core method development in this research does not involve LLMs as any important, original, or non-standard components.
- Please refer to our LLM policy (`https://neurips.cc/Conferences/2025/LLM`) for what should or should not be described.

