# OpenReview forum: "Multi-Actor Multi-Critic Deep Deterministic Reinforcement Learning with a Novel Q-Ensemble Method"
_NeurIPS.cc/2025/Conference — Submitted to NeurIPS 2025_

### Official Review · Reviewer_cqoe · 2025-06-26

**Clarity:** 2
**Significance:** 2
**Originality:** 3
**Rating:** 3
**Confidence:** 4

**Summary:**

This paper proposes the Multi-Actor Multi-Critic (MAMC) deep deterministic reinforcement learning method, which introduces a framework with multiple actors and multiple critics to improve both exploration and value estimation in reinforcement learning tasks. MAMC features an innovative selection strategy based on non-dominated sorting, evaluating actors using both skill (expected return) and creativity (diversity of critic outputs). It employs a quantile-based ensemble approach for value estimation to address common issues, such as overestimation and underestimation biases. Theoretical analysis demonstrates that MAMC ensures stable learning and bounded estimation error. Experiments on MuJoCo benchmarks show that MAMC outperforms several leading methods in terms of convergence speed and final returns, especially in challenging environments. The study’s contributions include improved ensemble estimators, a new multi-objective actor selection framework, theoretical justifications, and robust empirical results on complex continuous control problems

**Questions:**

Q1: MAMC involves simultaneously maintaining and updating multiple actors and critics, which requires considerable computational resources. Could the authors provide a more detailed breakdown of the runtime, memory usage, and scalability as the number of actors/critics increases, especially compared to baselines?
Q2: The paper acknowledges that stochastic actor-critic methods such as REDQ-SMR may outperform the deterministic-policy MAMC on the most challenging environments. Could the authors discuss in more detail the underlying reasons for this and whether extensions to stochastic or hybrid policy frameworks are feasible within the MAMC?
Q3: The non-dominated sorting based on skill and creativity for actor selection is novel, and ablation shows its benefit. Can you provide a more detailed analysis or visualization of how the balance between exploration and exploitation evolves over time and across environments?

**Ethical Concerns:**

["NO or VERY MINOR ethics concerns only"]

**Limitations:**

yes

**Paper Formatting Concerns:**

No major formatting problems were found.

**Quality:**

3

**Strengths And Weaknesses:**

Strengths:
1) Novel Architecture: The paper introduces a unique Multi-Actor Multi-Critic (MAMC) framework that incorporates multiple actors and critics with a sophisticated selection mechanism, which is innovative compared to prior actor-critic architectures.
2) Balanced Exploration and Exploitation: The use of a two-objective actor selection mechanism (skill and creativity factors) based on non-dominated sorting helps balance the trade-off between finding high-quality actions and maintaining diverse exploration, supporting robust learning.
3) Quantile-Based Ensemble Estimation: The quantile-based ensemble strategy for critic evaluation addresses common estimation biases (over- and underestimation), leading to more stable and accurate value estimates.

Weakness:
1) Scalability Concerns: Maintaining and training multiple actors and critics may lead to increased computational and memory demands, which could become a bottleneck for large-scale or resource-constrained applications.
2) Limited Stochastic Policy Comparison: While comparisons with both deterministic and stochastic methods are included, the authors acknowledge that stochastic approaches like REDQ-SMR can surpass MAMC in some environments, suggesting potential limitations of the deterministic framework.
3) Hyperparameter Sensitivity: The performance of MAMC can be sensitive to the setting of the quantile parameter q, and the best value appears environment-dependent, requiring tuning for different tasks.
4) It is recommended to further strengthen English writing, with a focus on the accurate use of verb tenses.

---

> ### Author Rebuttal · Authors · 2025-07-31
>
> We appreciate the reviewer to provide all the valuable comments for improving our study. Our rebuttal and responses are listed as follows.
>
> W1: We thank the reviewer for pointing out the issue of scalability. It is true that the consideration of multiple actors and multiple critics introduces extra computational cost. However, the proposed MAMC have potential benefit over other architectures in some circumstances. For instance, the MAMC is suitable for designing models which are to be deployed on end systems with constrained computational power and/or memory. The benefits over other architectures on such circumstance are three folds. First, the MAMC returns only a single actor so that the computational power for inference and memory requirement are the same as the architectures with a single actor. Second, the MAMC has the ability to leverage more computational power for training by increasing the number of actors. Third, the MAMC is a flexible architecture that it can accept any setting for the number of actors and the number of critics. We can take advantage of these benefits of MAMC on computationally powerful devices and deploy the lightweight and effective actor to resource-constrained applications. In addition, we can also lowering down the number of actors and critics for fine-tuning the actor for matching the resource requirement. As aforementioned, the MAMC is a flexible architecture that we can adjust the number of actors and critics to get a satisfactory computational cost.
>
> W2: We thank the reviewer for providing the concerns. Originally, this study, as our title, is to propose an MAMC architecture for deterministic method. It is just for a reference to provide the comparison of the deterministic MAMC to stochastic-based methods such as SAC-SMR and REDQ-SMR. In fact, the size of each actor network in the two stochastic-based method is larger than the size of those in the three deterministic-based methods, i.e., TD3-SMR, DARC-SMR, and our MAMC. With limited time during the rebuttal period, we integrate stochastic actor into our MAMC architecture, forming the stochastic MAMC (sMAMC), and conduct experiments to show the effectiveness of the MAMC architecture.
>
> The following table lists the average return for SAC-SMR, REDQ-SMR, and stochastic MAMC (sMAMC) on Hopper-v5, Walker2D-v5, Ant-v5, and Humanoid-v5. The results are over 10 trials except that the sMAMC on Humanoid-v5 are averaged over 6 trials due to the limited time period. The bold symbol indicates the highest average return. The sMAMC outperforms SAC-SMR and REDQ-SMR on Walker2d-v5 and Humanoid-5. On Hopper-v5, the three methods have comparable results. As for Ant-v5, the sMAMC excels SAC-SMR; however, the sMAMC is surpassed by REDQ-SMR at later stage. These results indicate that the sMAMC achieves better quality in comparison to SAC-SMR and REDQ-SMR, and this is able to verify that the MAMC architecture is also applicable for stochastic method. Please note that we do not test and tune any parameter for the sMAMC, and such performance can be regarded as a baseline performance.
>
> |||SAC-SMR|SAC-SMR|REDQ-SMR|REDQ-SMR|sMAMC|sMAMC|
> |:-|-:|-:|-:|-:|-:|-:|-:|
> |Env.|Steps|Mean|Std.|Mean|Std.|Mean|Std.|
> |Hopper-v5|50k|1898|1057|1794|1068|**2879**|514|
> ||100k|2732|1118|2993|667|**3181**|304|
> ||150k|**3340**|81|3211|691|3162|675|
> ||200k|3350|123|**3484**|108|3408|132|
> ||250k|3373|116|**3495**|128|3424|141|
> ||300k|**3394**|86|3335|605|3142|1104|
> |Walker2d-v5|50k|766|415|1160|663|**1903**|1212|
> ||100k|1975|938|2998|903|**3190**|1265|
> ||150k|3232|1171|3410|1318|**4171**|348|
> ||200k|3591|398|4065|1011|**4459**|352|
> ||250k|3601|845|4139|958|**4576**|296|
> ||300k|3832|1268|4431|648|**4615**|366|
> |Ant-v5|50k|405|135|884|312|**1380**|794|
> ||100k|536|134|2538|730|**2948**|918|
> ||150k|667|97|3554|620|**3683**|1123|
> ||200k|812|268|**4099**|1170|3917|1226|
> ||250k|1071|462|**4722**|804|4071|1072|
> ||300k|1419|859|**4924**|573|4314|1136|
> |Humanoid-v5|50k|555|55|672|112|**826**|282|
> ||100k|771|227|1390|921|**1739**|1074|
> ||150k|1326|1067|1995|886|**2175**|1374|
> ||200k|2089|1187|3207|1186|**3746**|1251|
> ||250k|3251|803|**4705**|719|4678|536|
> ||300k|4226|1064|4448|1321|**4939**|237|
>
> We also compare deterministic methods to stochastic methods to show the correctness of our claim, i.e., on the same or similar architecture, stochastic-based method is superior to deterministic-based method. The sMAMC outperforms dMAMC on the four test environments. Similar trend can be found in the comparisons of TD3-SMR and SAC-SMR that the SAC-SMR obtains higher average return than TD3-SMR does on Hopper-v5, Ant-v5, and Humanoid-v5, except that on Walker2d-v5 SAC-SMR is inferior to TD3-SMR. These results validate that the stochastic-based method is superior to the deterministic-based method on the SADC and MAMC architecture.
>
> ||TD3-SMR|TD3-SMR|SAC-SMR|SAC-SMR|dMAMC|dMAMC|sMAMC|sMAMC|
> |:-|-:|-:|-:|-:|-:|-:|-:|-:|
> |Env|Mean|Std.|Mean|Std.|Mean|Std.|Mean|Std.|
> |Hopper-v5|3134|885|**3394**|86|2928|909|**3142**|1104|
> |Walker2d-v5|**4189**|647|3832|1268|4210|585|**4615**|366|
> |Ant-v5|698|759|**1419**|859|3988|1534|**4314**|1136|
> |Humanoid-v5|1808|1470|**4226**|1064|4070|1119|**4939**|237|
>
> W3: We thank reviewer for giving the valuable comment on the sensitivity of hyperparameter. Section 6.5 analyzes the sensitivity of the quantile q. The results give two points for setting this hyperparameter. First, the setting of q is insensitive in a certain range; that is, the performance would NOT dramatically change when the q changes slightly. Second, the setting of q is more general and more effective for controlling the view to the environment. By setting a high value to q, the MAMC behaves more optimistic and performs well on the HalfCheetah-v5 environment, which is recognized as an environment requires optimistic critics in past studies. On the contrary, for the environment which requires less optimistic critics, e.g., the Walker2d-v5, lowering down q to 0.2 or 0.3 would lead to higher return. Noteworthily, it is limited (f.i. DARC and REDQ) or inapplicable (f.i. TD3 and SAC) to change the view to the environment for the compared methods. In general, when using the MAMC, we can just pick a robust setting of q such as 0.2 if we have no prior knowledge to environment; otherwise, we can lowering down the q to make the ensemble critics more pessimistic or enlarging the q to make the ensemble critics more optimistic, depending on the knowledge to the environment. In addition, the SOTA methods such as SAC-SMR, DARC-SMR and REDQ-SMR all have fine-tuned hyperparameter for the test environments. Yet, we use a single robust setting of quantile q=0.2 for conducting the experiments and making comparisons. This again reflects the effectiveness of the MAMC and the robustness of quantile method.
>
> W4: We thank the reviewer for mentioning this. We will improve the English writing and proofread carefully in the next revision.
>
> Q1: We thank the reviewer for asking the question. The memory consumption is proportional to the number of actors and critics so we focus on the analysis of time complexity. The following table compiles the time complexity of the test methods considering parallelization, where A, C, and M denote the number of actors, critics, and the sample multiple reuse (SMR) ratio. The time complexity is computed based on the number of forward inference and backward propagation for all actors and critics over a batch. Note that the last complexity for the MAMC is computed based on our parallel implementation. The MAMC requires more calls at forward inference than at backward propagation, but forward inference is way faster than backward propagation.
>
> |||TD3-SMR|DARC-SMR|SAC-SMR|REDQ-SMR|MAMC|MAMC|
> |:-:|:-:|:-:|:-:|:-:|:-:|:-:|:-:|
> |Type|Direction|||||w/o parallel|w/ parallel|
> |Actor|Forward|M+M/2=15|4M+2M=60|M+M=20|M+1=11|A+ACM+AM=1110|A+AM+AM=210|
> ||Backward|M/2=5|2M=20|M=10|1=1|AM=100|AM=100|
> |Critic|Forward|4M+M/2=45|2+10M+2M=122|4M+2M=60|2M+CM+C=130|AC+ACCM+CM+AM=10300|C+CM+CM+M=220|
> ||Backward|2M+M/2=25|2M+2M=40|2M+M=30|CM+C=110|CM+AM=200|CM+M=110|
>
> We also calculate the running time for the test methods. The following table compiles the mean execution time per thousand environmental steps in seconds for the six test methods on Hopper-v5 and Humanoid-v5 with NVIDIA RTX 4090. The running time of MAMC is linear to the number of actors and critics, by comparison with DARC-SMR and REDQ-SMR.
> ||TD3-SMR|DARC-SMR|SAC-SMR|REDQ-SMR|dMAMC|sMAMC|
> |:-|-:|-:|-:|-:|-:|-:|
> |Hopper-v5|15.81|40.14|30.52|73.95|208.97|290.96|
> |Humanoid-v5|16.86|40.38|32.63|78.76|210.26|294.54|
>
> Next we analyze the effect on the number of actors and critics. The results show that the number of actors has more impacts on the running time.
> ||2A10C|5A10C|10A10C|10A5C|10A2C|
> |:-|-:|-:|-:|-:|-:|
> |Humanoid-v5|182.04|188.75|210.26|134.23|88.99|
>
> Q2: The rebuttal is responded together in the rebuttal for W2.
>
> Q3: We thank the reviewer for appreciating the novelty. In Section 6.4, we look into the multi-objective actor selection given in 4.3. From Fig. 2, the results indicate that the average return of the actor selected by our design will come close to the one with highest average return out of the ten actors, which reflects the effectiveness of the selection mechanism. In addition, the differences between upper and lower bounds reflect the diversity among the actors, which provides an evidence that the selection from these actors may be explorative, if the corresponding return is high, or exploitative, if the corresponding return is low. The existence of explorative and exploitative actors, and the effectiveness of the multi-objective selection over single-objective selection verify that the proposed MAMC is capable of balancing the exploration and exploitation. We will add the detailed analysis in the next revision for improving the clarity.

---

> > ### Author Response · Authors · 2025-08-05
> > **Appreciation to the Reviewer and Initiation of Discussion**
> >
> > Dear reviewer,
> >
> > We highly appreciate your valuable comments to our study. We have addressed the issues on the aspects of novelty, technical contributions, effectiveness on stochastic-based methods, scalability, sensitivity, and deeper analysis of the proposed components. We have also responded to the questions mentioned in the comments, and clarified the descriptions in our study.
> >
> > Please feel free to let us know if there is any concern which has not yet been well-addressed or well-clarified enough.
> >
> > If you appreciate this study after our clarification, it would be of great help if you could update the ratings.
> >
> > We are looking forward to seeing your response for further discussion.

---

> > ### Comment · Reviewer_cqoe · 2025-08-06
> >
> > Thanks for answering my questions. Due to the novelty and weakness, I will maintain my score.

---

> > > ### Author Response · Authors · 2025-08-06
> > >
> > > We thank the reviewer for responding to our rebuttals. From the rebuttals, we have addressed the weakness and answered the questions metioned by the reviewer. Could you please provide the issues of novelty and weakness that we have not yet well-addressed?  It would be of great help for further clarification.

---

### Official Review · Reviewer_AYRp · 2025-07-02

**Clarity:** 2
**Significance:** 2
**Originality:** 2
**Rating:** 3
**Confidence:** 3

**Summary:**

The authors proposed a multi-actor based policy gradient method in this submission, in contrast to the single-actor methods in some previous work. The proposed method contains a few new features, mainly focusing on quantile based ensemble estimation for actor/critic evaluation and actor selection. Furthermore, the authors provided some theoretical analysis regarding variance reduction. The experimental results demonstrated the superior performance of the proposed method over a few baseline methods.

**Questions:**

* what did you mean by the ``a current manner'' in L47?
* what's the definition for `skill` and `creativity` in L174?
* what's the motivation of using median over quantile in Eq. (6)?

**Ethical Concerns:**

["NO or VERY MINOR ethics concerns only"]

**Final Justification:**

I will keep my score because (i) the theorems in Section 5 show the benefits from variance, but the results on the estimation errors look insufficient; (ii) the boundary between stochastic and deterministic nature of MAMC vs. Sunrise is also a bit blurry.

**Limitations:**

yes

**Quality:**

2

**Strengths And Weaknesses:**

## Strength:

Given that multiple-actor has been employed in previous methods, e.g., SUNRISE, the major contribution from this submission is maily on the quantile base ensemble estimation, which looks interesting and is somewhat justified in practice.


## Weakness:

1. Even though the proposed quantile based ensemble estimation approach looks interesting, some in-depth analysis is necessary to explain why it should work well. The authors may hint about its benefits in Section 4.3, but I was not able to identify them.

2. Given a few other related methods also belong to the multi-actor multi-critic family, why not compare against them in the experiments, e.g., SUNRISE?

3. There are a few typos and some claims may look incorrct, for which are summarized below.

   * In Eq. (2) and Eq. (7), shouldn't the sign in front of $\gamma$ be negative?
   * DQN is considered to be value based method so it should not be included in Table 1.

---

> ### Author Rebuttal · Authors · 2025-07-31
>
> We appreciate the reviewer to provide all the valuable comments for improving our study. Our rebuttal and responses are listed as follows.
>
> S: We thank the reviewer for appreciating the contribution on the quantile base ensemble estimation. To the best of our knowledge, the proposed MAMC architecture is the first multi-actor multi-critic method which decoupled the interaction of actors and critics. Existing methods such as DARC and SUNRISE are based on integration of multiple actor-critic architectures. However, the MAMC does NOT constrain the one-to-one mapping among all the actors and critics. We think this can also reflect the novelty of our study.
>
> W1: We thank the reviewer for giving the valuable comment. In this study, we have investigated the effectiveness of the proposed MAMC theoretically and empirically. Section 5 provides theoretical analysis for the MAMC, and the corresponding proofs are given in supplementary material. From the deduction, we provide evidences that the proposed MAMC is more stable, with less variance, than SAMC and SASC. We also show the required constraint for the claimed stability (cf. Eqs (18), (22), and (23)). We may consider combining the claims and the proofs in the manuscript for improving the clarity.
>
> In Section 6.2 and 6.3, we compare the MAMC to SOTA methods of different architectures through experiments on a set of well-known benchmark problems to show its effectiveness. The results show that the MAMC is competitive and performs nicely on more complicated environments such as Ant-v5 and Humanoid-v5. In Section 6.4, we look into the multi-objective actor selection given in 4.3. The results validate that the design is effective that the result is superior to single-objective selection and in most environments the selected actor by our design will come close to the one with highest average return out of the ten actors.
>
> Section 6.5 further analyzes the sensitivity of a key hyperparameter in the MAMC, i.e., the quantile q. The results give two points for setting this hyperparameter. First, the setting of q is insensitive in a certain range; that is, the performance would NOT dramatically change when the q changes slightly. Second, the setting of q is more general and more effective for controlling the view to the environment. By setting a high value to q, the MAMC behaves more optimistic and performs well on the HalfCheetah-v5 environment, which is recognized as an environment requires optimistic critics in past studies. On the contrary, for the environment which requires less optimistic critics, e.g., the Walker2d-v5, lowering down q to 0.2 or 0.3 would lead to higher return. Noteworthily, it is limited (f.i. DARC and REDQ) or inapplicable (f.i. TD3 and SAC) to change the view to the environment for the compared methods. In general, when using the MAMC, we can just pick a robust setting of q such as 0.2 if we have no prior knowledge to environment; otherwise, we can lower down the q to make the ensemble critics more pessimistic or enlarging the q to make the ensemble critics more optimistic, depending on the knowledge to the environment.
>
> W2: We thank the reviewer for pointing out this issue. Though SUNRISE also adopts multiple actors and multiple critics, it is an ensemble policy, which required the participant of all actors to decide when interacting to the environment. By contrast, the MAMC is to seek a good policy, represented by a single actor, by manipulating multiple actors. This study focused on the impact of ensemble critics, and thus we did not add SUNRISE into comparison. Another reason for not including SUNRISE is that it is a stochastic-based method, and this study mainly focused on deterministic-based methods. We will add the above explanation to the next revision for improving the clarity.
>
> W3: We thank the reviewer for pointing out the typos, we will correct them in the next revision.
>
> Q1: This is a typo, and the correct one is “a concurrent manner”. We apology to make reviewer confused, and will correct them in the next revision.
>
> Q2: We apology the organization for making reviewer confused. The mathematical definitions of skill factor and creativity factor are given in equations (9) and (10), respectively. The descriptions about the two factor are from L180 to L185. We will try to re-organize the presentation for avoiding this issue.
>
> Q3: We thank the reviewer for asking this good question. The quantile is for addressing the issue of estimation accuracy with multiple critics, whilst the median is to alleviate the influence of outliers from averaging. We will add the motivation in the next revision for improving the clarity.

---

> > ### Author Response · Authors · 2025-08-05
> > **Appreciation to the Reviewer and Initiation of Discussion**
> >
> > Dear reviewer,
> >
> > We highly appreciate your valuable comments to our study. We have addressed the issues on the aspects of novelty, technical contributions, component analysis, and sensitivity analysis. We have also responded to the questions mentioned in the comments, and clarified the descriptions and typos in our study.
> >
> > Please feel free to let us know if there is any concern which has not yet been well-addressed or well-clarified enough.
> >
> > If you appreciate this study after our clarification, it would be of great help if you could update the ratings.
> >
> > We are looking forward to seeing your response for further discussion.

---

### Official Review · Reviewer_EoQu · 2025-07-02

**Clarity:** 2
**Significance:** 2
**Originality:** 1
**Rating:** 2
**Confidence:** 4

**Summary:**

This paper proposes a novel multi-actor multi-critic (MAMC) deep deterministic reinforcement learning method with quantile-based ensemble estimation for addressing the issues of estimation accuracy. MAMC has three key contributions: the non-dominated sorting for actor selection to explore skill and creativity, a quantile-based ensemble method for evaluating actors and critics, and exploiting the actor based on the best skill factor. The authors provide a theoretical analysis to prove the learning stability and bounded estimation bias for the MAMC. Empirical studies on the MuJoCo benchmark validate the effectiveness of MAMC, demonstrating superior performance compared to state-of-the-art deep deterministic-based reinforcement learning methods.

**Questions:**

Q1: Why do you use quantiles?

Q2: Is MAMC sensitive to the number of actors/critics?

Q3: Why were you interested in the issues of estimation accuracy in the deterministic RL problem?

**Ethical Concerns:**

["NO or VERY MINOR ethics concerns only"]

**Limitations:**

See above comments.

**Quality:**

2

**Strengths And Weaknesses:**

**Strength:**
1. The MAMC method introduces a novel multi-actor multi-critic RL architecture that addresses the limitations of existing single-actor, single/multi-critic RL methods.
2. MAMC presents nice theoretical properties and the bounds of the estimation error.

**Weakness:**
1. MAMC is confined to a deterministic setting. Its performance on stochastic RL methods is unknown.
2. It is not clear whether the gained performances are from the increased capacity of the neural network or the algorithmic improvement of the RL method.
3. MAMC does not always show superior performance in Figure 1. Only 5 scenarios are not enough to evaluate the effectiveness of the proposed method.
4. We can see the improvement of performance in some MoJoCo scenarios. But it is not clear what the exact level of the estimation MAMC can gain in this paper.

---

> ### Author Rebuttal · Authors · 2025-07-31
>
> We appreciate the reviewer to provide all the valuable comments for improving our study. Our rebuttal and responses are listed as follows.
>
> W1: We thank the reviewer for giving the valuable comment. This study is to introduce a novel MAMC architecture for deterministic policy. We understand your concern, and thus we provide the results of MAMC with stochastic policy, and compare the performance to SAC-SMR and REDQ-SMR. We have to confess that all the parameters are inherited from the MAMC with deterministic policy without any fine-tuning, and thus the results can be regarded as a lower bound performance of MAMC with stochastic policy.
>
> The following table lists the average return for SAC-SMR, REDQ-SMR, and stochastic MAMC (sMAMC) on Hopper-v5, Walker2D-v5, Ant-v5, and Humanoid-v5. The results are over 10 trials except that the sMAMC on Humanoid-v5 are averaged over 6 trials due to the limited time period. The bold symbol indicates the highest average return. The sMAMC outperforms SAC-SMR and REDQ-SMR on Walker2d-v5 and Humanoid-5. On Hopper-v5, the three methods have comparable results. As for Ant-v5, the sMAMC excels SAC-SMR; however, the sMAMC is surpassed by REDQ-SMR at later stage. These results indicate that the sMAMC achieves better quality in comparison to SAC-SMR and REDQ-SMR, and this is able to verify that the MAMC architecture is also applicable for stochastic method. Please note that we do not test and tune any parameter for the sMAMC, and such performance can be regarded as a baseline performance.
>
> |||SAC-SMR|SAC-SMR|REDQ-SMR|REDQ-SMR|sMAMC|sMAMC|
> |:-|-:|-:|-:|-:|-:|-:|-:|
> |Env.|Steps|Mean|Std.|Mean|Std.|Mean|Std.|
> |Hopper-v5|50k|1898.45|1057.06|1794.59|1068.79|**2879.63**|514.24|
> ||100k|2732.37|1118.75|2993.67|667.81|**3181.85**|304.75|
> ||150k|**3340.24**|81.94|3211.21|691.66|3162.31|675.06|
> ||200k|3350.94|123.93|**3484.63**|108.70|3408.67|132.38|
> ||250k|3373.25|116.96|**3495.73**|128.73|3424.84|141.30|
> ||300k|**3394.94**|86.58|3335.22|605.02|3142.70|1104.95|
> |Walker2d-v5|50k|766.71|415.25|1160.81|663.82|**1903.29**|1212.18|
> ||100k|1975.80|938.52|2998.72|903.84|**3190.80**|1265.59|
> ||150k|3232.80|1171.17|3410.81|1318.77|**4171.03**|348.20|
> ||200k|3591.62|398.67|4065.27|1011.02|**4459.88**|352.89|
> ||250k|3601.82|845.44|4139.12|958.61|**4576.84**|296.60|
> ||300k|3832.68|1268.07|4431.88|648.25|**4615.38**|366.83|
> |Ant-v5|50k|405.19|135.33|884.81|312.45|**1380.41**|794.38|
> ||100k|536.32|134.85|2538.33|730.35|**2948.41**|918.43|
> ||150k|667.84|97.18|3554.10|620.07|**3683.04**|1123.91|
> ||200k|812.22|268.57|**4099.54**|1170.57|3917.82|1226.09|
> ||250k|1071.09|462.89|**4722.67**|804.89|4071.26|1072.67|
> ||300k|1419.33|859.70|**4924.73**|573.37|4314.32|1136.13|
> |Humanoid-v5|50k|555.31|55.86|672.37|112.15|**826.80**|282.93|
> ||100k|771.14|227.39|1390.48|921.64|**1739.38**|1074.71|
> ||150k|1326.47|1067.96|1995.01|886.75|**2175.34**|1374.90|
> ||200k|2089.61|1187.21|3207.24|1186.88|**3746.25**|1251.57|
> ||250k|3251.55|803.33|**4705.15**|719.53|4678.02|536.06|
> ||300k|4226.18|1064.51|4448.81|1321.48|**4939.16**|237.24|
>
> We also compare deterministic methods to stochastic methods to show the correctness of our claim, i.e., on the same or similar architecture, stochastic-based method is superior to deterministic-based method. The following table lists the average return for SAC-SMR, REDQ-SMR, and stochastic MAMC (sMAMC) on Hopper-v5, Walker2D-v5, Ant-v5, and Humanoid-v5. The results are over 10 trials except that the sMAMC on Humanoid-v5 are averaged over 6 trials due to the limited time period. The bold symbol indicates the highest average return. The sMAMC outperforms dMAMC on the four test environments. Similar trend can be found in the comparisons of TD3-SMR and SAC-SMR that the SAC-SMR obtains higher average return than TD3-SMR does on Hopper-v5, Ant-v5, and Humanoid-v5, except that on Walker2d-v5 SAC-SMR is inferior to TD3-SMR at later stage. These results validate that the stochastic-based method is superior to the deterministic-based method on the SADC and MAMC architecture.
>
> |||TD3-SMR|TD3-SMR|SAC-SMR|SAC-SMR|dMAMC|dMAMC|sMAMC|sMAMC|
> |:-|-:|-:|-:|-:|-:|-:|-:|-:|-:|
> |Env.|Steps|Mean|Std.|Mean|Std.|Mean|Std.|Mean|Std.|
> |Hopper-v5|50k|1689.85|778.66|**1898.45**|1057.06|1743.73|1229.35|**2879.63**|514.24|
> ||100k|**2879.33**|506.97|2732.37|1118.75|2352.92|1062.76|**3181.85**|304.75|
> ||150k|3304.31|108.52|**3340.24**|81.94|2571.26|1018.23|**3162.31**|675.06|
> ||200k|2852.91|743.80|**3350.94**|123.93|2173.23|1241.61|**3408.67**|132.38|
> ||250k|3175.55|527.69|**3373.25**|116.96|2658.50|1107.61|**3424.84**|141.30|
> ||300k|3134.43|885.81|**3394.94**|86.58|2928.80|909.29|**3142.70**|1104.95|
> |Walker2d-v5|50k|445.59|126.72|**766.71**|415.25|1527.81|955.83|**1903.29**|1212.18|
> ||100k|1717.87|1043.55|**1975.80**|938.52|2949.83|949.37|**3190.80**|1265.59|
> ||150k|2950.19|954.83|**3232.80**|1171.17|3926.33|459.86|**4171.03**|348.20|
> ||200k|**3961.21**|672.74|3591.62|398.67|3897.89|681.05|**4459.88**|352.89|
> ||250k|**3653.41**|1124.02|3601.82|845.44|4194.85|456.38|**4576.84**|296.60|
> ||300k|**4189.75**|647.20|3832.68|1268.07|4210.10|585.83|**4615.38**|366.83|
> |Ant-v5|50k|99.92|46.79|**405.19**|135.33|1104.34|690.48|**1380.41**|794.38|
> ||100k|53.03|92.75|**536.32**|134.85|2432.91|1300.14|**2948.41**|918.43|
> ||150k|156.75|287.19|**667.84**|97.18|2854.80|1489.39|**3683.04**|1123.91|
> ||200k|360.51|674.02|**812.22**|268.57|3151.39|1758.29|**3917.82**|1226.09|
> ||250k|536.50|920.19|**1071.09**|462.89|3599.31|1652.70|**4071.26**|1072.67|
> ||300k|698.28|759.79|**1419.33**|859.70|3988.73|1534.31|**4314.32**|1136.13|
> |Humanoid-v5|50k|444.40|70.35|**555.31**|55.86|692.69|176.12|**826.80**|282.93|
> ||100k|596.98|107.70|**771.14**|227.39|1612.54|1409.57|**1739.38**|1074.71|
> ||150k|694.05|159.79|**1326.47**|1067.96|**2615.86**|1566.08|2175.34|1374.90|
> ||200k|1077.55|664.35|**2089.61**|1187.21|3068.25|1446.61|**3746.25**|1251.57|
> ||250k|1531.61|1173.12|**3251.55**|803.33|3581.72|1254.78|**4678.02**|536.06|
> ||300k|1808.36|1470.58|**4226.18**|1064.51|4070.58|1119.40|**4939.16**|237.24|
>
> W2: We thank the reviewer for offering the concern. About increasing the number of critics, we have shown that the MAMC with ten critics is a robust setting, and leveraging more or less number of critics may result in performance deterioration. The same trend can be found in adjusting the number of actors. Also, past studies such as REDQ have also shown that the performance would not continually increase as the number of critics increases. This observation verifies that it is essential to have an appropriate setting for the number of actors and critics. Moreover, the proposed MAMC have potential benefit over other architectures in some circumstances. For instance, the MAMC is suitable for designing models which are to be deployed on end systems with constrained computational power and/or memory. The benefits over other architectures on such circumstance are three folds. First, the MAMC returns only a single actor so that the computational power for inference and memory requirement are the same as the architectures with a single actor. Second, the MAMC has the ability to leverage more computational power for training by increasing the number of actors. Third, the MAMC is a flexible architecture that it can accept any setting for the number of actors and the number of critics. Furthermore, the proof section provides theoretical foundation for the effectiveness of the proposed MAMC method. From deduction, we provide evidences that the proposed MAMC is more stable, with less variance, than SAMC and SASC. The above observations can validate the improvements are from the proposed MAMC architecture.
>
> W3: We thank the reviewer for pointing out the issue. About the effectiveness, comparing to TD3 and DARC the MAMC performs best on the most complicated two environments, i.e., Ant-v5 and Humanoid-v5, and the MAMC is comparable to DARC on simpler Walker2D-v5 and HalfCheetah-v5. These results can validate the proposed MAMC is effective, especially on complicated environments. For the number of scenarios, we understand the reviewer's concern about the number of tested environments. Nevertheless, these environments are well-known and have been widely applied as benchmark problems for comparing the performance of RL methods. Moreover, we also compare the stochastic MAMC (sMAMC) to the two stochastic-based methods in above table, and the results verify that the sMAMC is effective.
>
> W4: We thank the reviewer for offering the concern. From the experiments, the MAMC can achieve nice performance with efficient inference on complicated environments such as Ant-v5 and Humanoid-v5. The results validate the effectiveness and efficiency of the proposed MAMC.
>
> Q1: The quantile for multiple critics is a generalization of existing strategies such as min, max, average, and so on, and it can prevent the influence from outliers for increasing the robustness.
>
> Q2: For the number of actors/critics, a proper setting is necessary. We suggest using ten actors and ten critics since such setting is more robust than all the other settings (cf. Figure 5 in supplementary material).
>
> Q3: Estimation accuracy is an essential issue for designing effective RL methods, and thus this study aims at developing a novel architecture, the MAMC, which is able to address the issue of estimation accuracy and the issue of balancing exploration and exploitation. The former is addressed by the quantile for multiple critics, whilst the latter is addressed by the selection among multiple actors based on their creativity factor and skill factor through a multi-objective comparator, i.e., the non-dominated sorting and crowding distance.

---

> > ### Comment · Area_Chair_nZ4L · 2025-08-06
> >
> > Dear reviewer EoQu,
> >
> > Could you please respond to authors' rebuttal as soon as possible?
> >
> > Thank you!
> > AC

---

> ### Author Response · Authors · 2025-08-05
> **Appreciation to the Reviewer and Initiation of Discussion**
>
> Dear reviewer,
>
> We highly appreciate your valuable comments to our study. We have addressed the issues on the aspects of novelty, technical contributions, effectiveness, and sensitivity. We have also responded to the questions, especially to the motivation and novelty, mentioned in the comments, and clarified the descriptions in our study.
>
> Please feel free to let us know if there is any concern which has not yet been well-addressed or well-clarified enough.
>
> We also list two questions to discuss.
> 1. Our rebuttal has clarified the issue of originality of this study. Do you have any further concerns about the novelty of this study?
> 2. Regarding weakness 3, we have testified five scenarios in the MuJoCo benchmark suite, which are commonly adopted in RL society. Do you have any suggestion to benchmarking the test methods for improving the significance of our study? This could be much helpful for strengthening our future study.
>
> If you appreciate this study after our clarification, it would be of great help if you could update the ratings.
>
> We are looking forward to seeing your response for further discussion.

---

> ### Comment · Reviewer_EoQu · 2025-08-07
> **Response to authors**
>
> Thanks for the additional experiments. Given the technical contributions of this paper, I keep the score.

---

> > ### Author Response · Authors · 2025-08-07
> >
> > We thank the reviewer for responding to our rebuttals. From the rebuttals, we have addressed the weakness and answered the questions metioned by the reviewer. Could you please provide the details of the concerns about technical contributions that we have not yet well-addressed? It would be of great help for further clarification.
> >
> > We also list two questions to discuss.
> > 1. Our rebuttal has clarified the issue of originality of this study. According to your current rating of originality, could you please elaborate your remained concerns?
> > 2. As the question of significance in weakness 3, we have testified five scenarios in the MuJoCo benchmark suite, which are commonly adopted in RL society. Could you please provide substantial suggestion to benchmarking the test methods which can meet the requirements of significance?

---

### Official Review · Reviewer_Ung8 · 2025-07-03

**Clarity:** 3
**Significance:** 2
**Originality:** 2
**Rating:** 3
**Confidence:** 5

**Summary:**

This paper proposes a novel multi-actor multi-critic deep deterministic reinforcement learning method (MAMC), aiming to address the issues of insufficient exploration and instability in traditional reinforcement learning methods. The method combines multiple actors and critics, introducing an actor selection mechanism based on skill and creativity factors, a quantile-based ensemble Q-value estimation strategy, and the exploitation of the actor with the best skill factor, which effectively improves stability and policy performance during the learning process. The paper demonstrates the advantages of the proposed method in terms of estimation bias control and learning stability.

Through experimental validation, MAMC performs excellently across multiple MuJoCo environments, outperforming existing deep deterministic reinforcement learning methods, particularly in the early and mid-training stages, where the effects are more pronounced. The experimental results also show that the actor selection mechanism, based on different policy qualities, can significantly enhance the model's generalization ability. This study provides an effective improvement to reinforcement learning methods with a multi-actor multi-critic structure.

**Questions:**

See above.

**Ethical Concerns:**

["NO or VERY MINOR ethics concerns only"]

**Final Justification:**

This work proposes the MAMC method, which performs integrated optimization based on the multi-actor multi-critic architecture, enhancing the stability of policy evaluation and the effectiveness of exploration. However, the innovation mainly lies in architectural integration, lacking theoretical breakthroughs and representing an incremental improvement. In addition, the multi-actor multi-critic design introduces significant computational cost.

Overall, the method has limited innovation and incurs high training time overhead. After considering the response of the authors and the discussions from other reviewers, I want to keep or reduce my score. Based on these obvious weeknesses, I recommend rejecting the paper.

**Limitations:**

Yes.

**Paper Formatting Concerns:**

None.

**Quality:**

3

**Strengths And Weaknesses:**

Strengths:
1) This paper proposes a novel multi-actor multi-critic (MAMC) deep deterministic reinforcement learning architecture that effectively combines multiple actors and critics, with no fixed pairings between them. Unlike traditional single actor-critic structures, the MAMC architecture provides greater flexibility by using multiple critics to evaluate the policy, while multiple actors participate in exploring the policy, thus effectively enhancing the diversity of the exploration process and the accuracy of the evaluation process.
2) The experimental design is reasonable and convincing. The authors test the proposed method across multiple standard MuJoCo environments, including Ant, Half-Cheetah, Hopper, and Humanoid, to comprehensively validate its effectiveness. Moreover, the paper compares the method with several mainstream reinforcement learning baseline methods (e.g., REDQ). The results show that MAMC significantly outperforms the baseline methods in various environments, further validating the effectiveness of the proposed approach.

Weaknesses:
1) The paper introduces a novel multi-actor multi-critic structure, but the MAMC method is mainly an extension and integrative optimization of the existing actor-critic structure. The primary improvement lies in enhancing the system-level integration, which improves the stability and accuracy of policy evaluation and exploration. However, this improvement is more focused on the combination and optimization of existing architectures rather than introducing a fundamentally new theoretical mechanism or paradigm, making the innovation incremental.
2) The proposed method uses 10 actors and 10 critics simultaneously. Although this design improves the effectiveness of exploration and evaluation, it may also incur high computational and training costs. While the paper briefly mentions the hardware configuration, it does not provide a detailed discussion of the computational burden introduced by the multi-actor and multi-critic design, nor does it evaluate the cost-effectiveness compared to other baseline methods.
3) According to the experimental curves, MAMC shows stable performance in the later stages of training, and in some cases, it is slightly inferior to REDQ. This indicates that the long-term performance advantage of the method is not prominent. The performance degradation in later training stages may be related to sample reuse or the multi-actor mechanism. The design of multiple actors might cause instability in policy updates during long-term training, especially when dealing with multi-batch samples and recursive training, leading to potential policy degradation or learning instability. Whether this issue is due to the frequency of policy updates or interactions between actors requires further analysis.

---

> ### Author Rebuttal · Authors · 2025-07-31
>
> We appreciate the reviewer to provide all the valuable comments for improving our study. Our rebuttal and responses are listed as follows.
>
> W1: We thank the reviewer for giving this comment. Actually, the MAMC is a framework with novel architecture that any design of actor and critic can be integrated into it. The MAMC considers quantile of multiple critics, which is a generalization of existing strategies such as min, max, average, and so on. In addition, the MAMC introduces the creativity factor and skill factor to assess actors, and adopts multi-objective comparator for balancing exploration and exploitation without the necessity of setting weights. These are fundamentally new mechanisms with theoretical foundations.
>
> W2: We thank the reviewer to point out the issue at computational cost. It is true that the consideration of multiple actors and multiple critics introduces extra computational cost. However, the proposed MAMC have potential benefit over other architectures in some circumstances. For instance, the MAMC is suitable for designing models which are to be deployed on end systems with constrained computational power and/or memory. The benefits over other architectures on such circumstance are three folds. First, the MAMC returns only a single actor so that the computational power for inference and memory requirement are the same as the architectures with a single actor. Second, the MAMC has the ability to leverage more computational power for training by increasing the number of actors. Third, the MAMC is a flexible architecture that it can accept any setting for the number of actors and the number of critics.
>
> Originally, this study, as our title, is to propose an MAMC architecture for deterministic method. It is just for a reference to provide the comparison of the deterministic MAMC to stochastic-based methods such as SAC-SMR and REDQ-SMR. In fact, the size of each actor network in the two stochastic-based method is larger than the size of those in the three deterministic-based methods, i.e., TD3-SMR, DARC-SMR, and our MAMC. With limited time during the rebuttal period, we integrate stochastic actor into our MAMC architecture, forming the stochastic MAMC (sMAMC), and conduct experiments to show the effectiveness of the MAMC architecture.
>
> The following table compiles the time complexity of the test methods considering parallelization, where A, C, and M denote the number of actors, critics, and the sample multiple reuse (SMR) ratio. The time complexity is computed based on the number of forward inference and backward propagation for all actors and critics over a batch. Note that the last complexity for the MAMC is computed based on our parallel implementation. The MAMC requires more calls at forward inference than at backward propagation, but forward inference is way faster than backward propagation.
>
> |||TD3-SMR|DARC-SMR|SAC-SMR|REDQ-SMR|MAMC|MAMC|
> |:-:|:-:|:-:|:-:|:-:|:-:|:-:|:-:|
> |Type|Direction|||||w/o parallel|w/ parallel|
> |Actor|Forward|M+M/2=15|4M+2M=60|M+M=20|M+1=11|A+ACM+AM=1110|A+AM+AM=210|
> ||Backward|M/2=5|2M=20|M=10|1=1|AM=100|AM=100|
> |Critic|Forward|4M+M/2=45|2+10M+2M=122|4M+2M=60|2M+CM+C=130|AC+ACCM+CM+AM=10300|C+CM+CM+M=220|
> ||Backward|2M+M/2=25|2M+2M=40|2M+M=30|CM+C=110|CM+AM=200|CM+M=110|
>
> We also calculate the running time for the test methods. The following table compiles the mean execution time per thousand environmental steps in seconds for the six test methods on Hopper-v5 and Humanoid-v5 with NVIDIA RTX 4090. The running time of MAMC is linear to the number of actors and critics, by comparison with DARC-SMR and REDQ-SMR.
> ||TD3-SMR|DARC-SMR|SAC-SMR|REDQ-SMR|dMAMC|sMAMC|
> |:--|--:|--:|--:|--:|--:|--:|
> |Hopper-v5|15.81|40.14|30.52|73.95|208.97|290.96|
> |Humanoid-v5|16.86|40.38|32.63|78.76|210.26|294.54|
>
> Next we analyze the effect on the number of actors and critics. The results show that the number of actors has more impacts on the running time.
> ||2A10C|5A10C|10A10C|10A5C|10A2C|
> |:--|--:|--:|--:|--:|--:|
> |Humanoid-v5|182.04|188.75|210.26|134.23|88.99|
>
> W3: We thank the reviewer for giving the valuable comment. It is true that the consideration of multiple actors introduces interaction among them. We analyze this impact in two experiments. In comparisons to TD3 (with a single actor) and DARC (with two actors), the MAMC (with ten actors) is effective on more complicated environments (Ant-v5 and Humanoid-v5). In comparison to MAMC with different number actors (cf. Figure 5 in supplementary material), the use of ten actors is more robust than MAMC with fewer number of actors. From the above observations, we can verify the effectiveness of using multiple actors.
>
> The following table lists the average return for SAC-SMR, REDQ-SMR, and stochastic MAMC (sMAMC) on Hopper-v5, Walker2D-v5, Ant-v5, and Humanoid-v5. The results are over 10 trials except that the sMAMC on Humanoid-v5 are averaged over 6 trials due to the limited time period. The bold symbol indicates the highest average return. The sMAMC outperforms SAC-SMR and REDQ-SMR on Walker2d-v5 and Humanoid-5. On Hopper-v5, the three methods have comparable results. As for Ant-v5, the sMAMC excels SAC-SMR; however, the sMAMC is surpassed by REDQ-SMR at later stage. These results indicate that the sMAMC achieves better quality in comparison to SAC-SMR and REDQ-SMR, and this is able to verify that the MAMC architecture is also applicable for stochastic method. Please note that we do not test and tune any parameter for the sMAMC, and such performance can be regarded as a baseline performance.
>
> |||SAC-SMR|SAC-SMR|REDQ-SMR|REDQ-SMR|sMAMC|sMAMC|
> |:-|-:|-:|-:|-:|-:|-:|-:|
> |Env.|Steps|Mean|Std.|Mean|Std.|Mean|Std.|
> |Hopper-v5|50k|1898.45|1057.06|1794.59|1068.79|**2879.63**|514.24|
> ||100k|2732.37|1118.75|2993.67|667.81|**3181.85**|304.75|
> ||150k|**3340.24**|81.94|3211.21|691.66|3162.31|675.06|
> ||200k|3350.94|123.93|**3484.63**|108.70|3408.67|132.38|
> ||250k|3373.25|116.96|**3495.73**|128.73|3424.84|141.30|
> ||300k|**3394.94**|86.58|3335.22|605.02|3142.70|1104.95|
> |Walker2d-v5|50k|766.71|415.25|1160.81|663.82|**1903.29**|1212.18|
> ||100k|1975.80|938.52|2998.72|903.84|**3190.80**|1265.59|
> ||150k|3232.80|1171.17|3410.81|1318.77|**4171.03**|348.20|
> ||200k|3591.62|398.67|4065.27|1011.02|**4459.88**|352.89|
> ||250k|3601.82|845.44|4139.12|958.61|**4576.84**|296.60|
> ||300k|3832.68|1268.07|4431.88|648.25|**4615.38**|366.83|
> |Ant-v5|50k|405.19|135.33|884.81|312.45|**1380.41**|794.38|
> ||100k|536.32|134.85|2538.33|730.35|**2948.41**|918.43|
> ||150k|667.84|97.18|3554.10|620.07|**3683.04**|1123.91|
> ||200k|812.22|268.57|**4099.54**|1170.57|3917.82|1226.09|
> ||250k|1071.09|462.89|**4722.67**|804.89|4071.26|1072.67|
> ||300k|1419.33|859.70|**4924.73**|573.37|4314.32|1136.13|
> |Humanoid-v5|50k|555.31|55.86|672.37|112.15|**826.80**|282.93|
> ||100k|771.14|227.39|1390.48|921.64|**1739.38**|1074.71|
> ||150k|1326.47|1067.96|1995.01|886.75|**2175.34**|1374.90|
> ||200k|2089.61|1187.21|3207.24|1186.88|**3746.25**|1251.57|
> ||250k|3251.55|803.33|**4705.15**|719.53|4678.02|536.06|
> ||300k|4226.18|1064.51|4448.81|1321.48|**4939.16**|237.24|

---

> > ### Comment · Reviewer_Ung8 · 2025-08-05
> >
> > Thank you for your reply. I will keep my score unchanged.

---

> > > ### Author Response · Authors · 2025-08-05
> > > **Appreciation to the Reviewer and Initiation of Discussion**
> > >
> > > Dear reviewer,
> > >
> > > Thank you very much for responding to our rebuttal.
> > >
> > > We highly appreciate your valuable comments to our study. We have addressed the issues on the aspects of novelty, technical contributions, scalability, and effectiveness on stochastic-based methods. We have also responded to the questions mentioned in the comments, and clarified the descriptions in our study.
> > >
> > > Would you please let us know if there is any concern which has not yet been well-addressed or well-clarified enough?
> > >
> > > If you appreciate this study after our clarification, it would be of great help if you could update the ratings.
> > >
> > > We are looking forward to seeing your response for further discussion.

---

### Official Review · Reviewer_13ae · 2025-07-03

**Clarity:** 2
**Significance:** 2
**Originality:** 2
**Rating:** 3
**Confidence:** 3

**Summary:**

This work proposes a novel multi-actor, multi-critic algorithm to improve performance compared to single-actor or single-critic approaches. The authors design a quantile-based ensemble value function to evaluate the performance of different policies (actors). In addition, the algorithm introduces a creativity factor and a skill factor to guide exploration. The paper provides theoretical analysis to support the effectiveness of the method and includes diverse simulation results and evaluation metrics to validate the proposed algorithm.

**Questions:**

1. How should the weights of the creativity factor and skill factor be selected in practice to balance exploration and exploitation, particularly for challenging tasks with high variance? For more difficult environments, would reducing the weight of the creativity factor help stabilize learning?

2. Could the authors provide results using different weight settings for the creativity and skill factors in the Ant-v5 and Humanoid-v5 environments? This would help demonstrate the sensitivity of the method to these parameters.

3. In Equation (6), first line, why is the mid quantile used instead of the average across critics? Does the mid quantile offer specific advantages, such as robustness to outliers or overestimation?

**Ethical Concerns:**

["NO or VERY MINOR ethics concerns only"]

**Final Justification:**

Based on the rebuttal and the information provided, I believe the level of novelty in this work does not justify an increased score. First, while the MAMC algorithm shows some performance improvement, it is not significantly better than other algorithms, yet it incurs substantially higher computational cost. Second, from a practical perspective, I find the design of the MAMC algorithm to be unreasonable. Therefore, I will maintain my original score.

**Limitations:**

Yes

**Quality:**

3

**Strengths And Weaknesses:**

***Strengths***
1.  The design of the Q-ensemble method is novel and generalizable. This metric has potential applications in a wide range of reinforcement learning problems.

2. The simulation section provides sufficient and persuasive experiments that effectively validate the proposed method.

***Weaknesses***
1. The theoretical proof section is somewhat trivial and lacks depth.

2. The MAMC algorithm significantly increases computational cost, which may limit its practicality—especially in large-scale models.

---

> ### Author Rebuttal · Authors · 2025-07-31
>
> We appreciate the reviewer to provide all the valuable comments for improving our study. Our rebuttal and responses are listed as follows.
>
> S: We thank the reviewer fo appreciating our novelty and contribution. Originally, this study, as our title, is to propose an MAMC architecture for deterministic method. It is just for a reference to provide the comparison of the deterministic MAMC to stochastic-based methods such as SAC-SMR and REDQ-SMR. In fact, the size of each actor network in the two stochastic-based method is larger than the size of those in the three deterministic-based methods, i.e., TD3-SMR, DARC-SMR, and our MAMC. With limited time during the rebuttal period, we integrate stochastic actor into our MAMC architecture, forming the stochastic MAMC (sMAMC), and conduct experiments to show the effectiveness of the MAMC architecture.
>
> The following table lists the average return for SAC-SMR, REDQ-SMR, and stochastic MAMC (sMAMC) on Hopper-v5, Walker2D-v5, Ant-v5, and Humanoid-v5. The results are over 10 trials except that the sMAMC on Humanoid-v5 are averaged over 6 trials due to the limited time period. The bold symbol indicates the highest average return. The sMAMC outperforms SAC-SMR and REDQ-SMR on Walker2d-v5 and Humanoid-5. On Hopper-v5, the three methods have comparable results. As for Ant-v5, the sMAMC excels SAC-SMR; however, the sMAMC is surpassed by REDQ-SMR at later stage. These results indicate that the sMAMC achieves better quality in comparison to SAC-SMR and REDQ-SMR, and this is able to verify that the MAMC architecture is also applicable for stochastic method. Please note that we do not test and tune any parameter for the sMAMC, and such performance can be regarded as a baseline performance.
>
> |||SAC-SMR|SAC-SMR|REDQ-SMR|REDQ-SMR|sMAMC|sMAMC|
> |:-|-:|-:|-:|-:|-:|-:|-:|
> |Env.|Steps|Mean|Std.|Mean|Std.|Mean|Std.|
> |Hopper-v5|50k|1898.45|1057.06|1794.59|1068.79|**2879.63**|514.24|
> ||100k|2732.37|1118.75|2993.67|667.81|**3181.85**|304.75|
> ||150k|**3340.24**|81.94|3211.21|691.66|3162.31|675.06|
> ||200k|3350.94|123.93|**3484.63**|108.70|3408.67|132.38|
> ||250k|3373.25|116.96|**3495.73**|128.73|3424.84|141.30|
> ||300k|**3394.94**|86.58|3335.22|605.02|3142.70|1104.95|
> |Walker2d-v5|50k|766.71|415.25|1160.81|663.82|**1903.29**|1212.18|
> ||100k|1975.80|938.52|2998.72|903.84|**3190.80**|1265.59|
> ||150k|3232.80|1171.17|3410.81|1318.77|**4171.03**|348.20|
> ||200k|3591.62|398.67|4065.27|1011.02|**4459.88**|352.89|
> ||250k|3601.82|845.44|4139.12|958.61|**4576.84**|296.60|
> ||300k|3832.68|1268.07|4431.88|648.25|**4615.38**|366.83|
> |Ant-v5|50k|405.19|135.33|884.81|312.45|**1380.41**|794.38|
> ||100k|536.32|134.85|2538.33|730.35|**2948.41**|918.43|
> ||150k|667.84|97.18|3554.10|620.07|**3683.04**|1123.91|
> ||200k|812.22|268.57|**4099.54**|1170.57|3917.82|1226.09|
> ||250k|1071.09|462.89|**4722.67**|804.89|4071.26|1072.67|
> ||300k|1419.33|859.70|**4924.73**|573.37|4314.32|1136.13|
> |Humanoid-v5|50k|555.31|55.86|672.37|112.15|**826.80**|282.93|
> ||100k|771.14|227.39|1390.48|921.64|**1739.38**|1074.71|
> ||150k|1326.47|1067.96|1995.01|886.75|**2175.34**|1374.90|
> ||200k|2089.61|1187.21|3207.24|1186.88|**3746.25**|1251.57|
> ||250k|3251.55|803.33|**4705.15**|719.53|4678.02|536.06|
> ||300k|4226.18|1064.51|4448.81|1321.48|**4939.16**|237.24|
>
> W1: We thank the reviewer for pointing out the issue. The proof section is to provide some theoretical foundation for the effectiveness of the proposed MAMC method. Due to space limitation, the complete proof is provided in the supplementary material. From deduction, this section provides evidences that the proposed MAMC is more stable, with less variance, than SAMC, SADC, and SASC. We also show the required constraint for the claimed stability (cf. Eqs (18), (22), and (23)). We think the problem comes from the separation of claims and proofs, and we will combine it in the revision for improving the clarity.
>
> W2: We thank the reviewer for giving the valuable comment. It is true that the consideration of multiple actors and multiple critics introduces extra computational cost. However, the proposed MAMC have potential benefit over other architectures in some circumstances. For instance, the MAMC is suitable for designing models which are to be deployed on end systems with constrained computational power and/or memory. The benefits over other architectures on such circumstance are three folds. First, the MAMC returns only a single actor so that the computational power for inference and memory requirement are the same as the architectures with a single actor. Second, the MAMC has the ability to leverage more computational power for training by increasing the number of actors. Third, the MAMC is a flexible architecture that it can accept any setting for the number of actors and the number of critics.
>
> The memory consumption is proportional to the number of actors and critics so we focus on the analysis of time complexity.  The following table compiles the time complexity of the test methods considering parallelization, where A, C, and M denote the number of actors, critics, and the sample multiple reuse (SMR) ratio. The time complexity is computed based on the number of forward inference and backward propagation for all actors and critics over a batch. Note that the last complexity for the MAMC is computed based on our parallel implementation. The MAMC requires more calls at forward inference than at backward propagation, but forward inference is way faster than backward propagation.
>
> |||TD3-SMR|DARC-SMR|SAC-SMR|REDQ-SMR|MAMC|MAMC|
> |:-:|:-:|:-:|:-:|:-:|:-:|:-:|:-:|
> |Type|Direction|||||w/o parallel|w/ parallel|
> |Actor|Forward|M+M/2=15|4M+2M=60|M+M=20|M+1=11|A+ACM+AM=1110|A+AM+AM=210|
> ||Backward|M/2=5|2M=20|M=10|1=1|AM=100|AM=100|
> |Critic|Forward|4M+M/2=45|2+10M+2M=122|4M+2M=60|2M+CM+C=130|AC+ACCM+CM+AM=10300|C+CM+CM+M=220|
> ||Backward|2M+M/2=25|2M+2M=40|2M+M=30|CM+C=110|CM+AM=200|CM+M=110|
>
> We also calculate the running time for the test methods. The following table compiles the mean execution time per thousand environmental steps in seconds for the six test methods on Hopper-v5 and Humanoid-v5 with NVIDIA RTX 4090. The running time of MAMC is linear to the number of actors and critics, by comparison with DARC-SMR and REDQ-SMR.
> ||TD3-SMR|DARC-SMR|SAC-SMR|REDQ-SMR|dMAMC|sMAMC|
> |:--|--:|--:|--:|--:|--:|--:|
> |Hopper-v5|15.81|40.14|30.52|73.95|208.97|290.96|
> |Humanoid-v5|16.86|40.38|32.63|78.76|210.26|294.54|
>
> Next we analyze the effect on the number of actors and critics. The results show that the number of actors has more impacts on the running time.
> ||2A10C|5A10C|10A10C|10A5C|10A2C|
> |:--|--:|--:|--:|--:|--:|
> |Humanoid-v5|182.04|188.75|210.26|134.23|88.99|
>
> Q1: We thank the review for asking about issue of setting weights for the two factors. To prevent from setting the weights for creativity factor and skill factor, the MAMC resolves the comparisons using a well-known multi-objective comparator, i.e., the non-dominated sorting and crowding distance. For balancing the exploration and exploitation, a random actor is picked from the non-dominated solutions (actors) with respect to the two objectives.
>
> Q2: From the rebuttal of Q1, it is no need for the MAMC to set the weights for the two objectives.
>
> Q3: We thank the reviewer for asking this question. Indeed, we use median rather than average to prevent the influence from outliers for increasing the robustness.

---

> > ### Author Response · Authors · 2025-08-05
> > **Appreciation to the Reviewer and Initiation of Discussion**
> >
> > Dear reviewer,
> >
> > We highly appreciate your valuable comments to our study. We have addressed the issues on the aspects of novelty, technical contributions, theoretical foundation, effectiveness on stochastic-based methods, and scalability. We have also responded to the questions mentioned in the comments, and clarified the descriptions in our study.
> >
> > Please feel free to let us know if there is any concern which has not yet been well-addressed or well-clarified enough.
> >
> > If you appreciate this study after our clarification, it would be of great help if you could update the ratings.
> >
> > We are looking forward to seeing your response for further discussion.

---

> > ### Comment · Area_Chair_nZ4L · 2025-08-06
> >
> > Dear reviewer 13ae,
> >
> > Could you please respond to authors' rebuttals as soon as possible?
> >
> > Thank you!
> > AC

---

> > ### Comment · Reviewer_13ae · 2025-08-06
> >
> > Thank you for the authors' response. My questions have been addressed. As I initially noted, the MAMC algorithm demonstrates better performance in most environments, but this comes at the cost of significantly increased computational overhead. From the response, I understand that the proposed structure has certain applications. However, based on the rebuttal and the information provided, I believe the level of novelty in this work does not warrant an increased score. Therefore, I will maintain my original rating.

---

> > > ### Author Response · Authors · 2025-08-06
> > >
> > > We thank the reviewer for responding to our rebuttal. We would like to emphasize that the cost at interaction to environment is assumed to be the most expensive part, and thus the proportion of training computational cost significantly decreases as the cost at interaction increases. Therefore, most experimental results are compared under the same number of environmental steps for a fair comparison. We have shown the benefits of the proposed method compared to the baselines under the same number of environmental steps.
> > >
> > > In addition, most existing methods such as TD3, SAC, DARC and REDQ are difficult to scale up in terms of the number of actors and/or critics. That is, these algorithms encounter challenges when attempting to add more actors or critics. It is also hard to simply increase the network size for improving the performance as the convergence is not guaranteed under a fixed computational budget. As a result, it is challenging to design experiments that vary in architecture while maintaining comparable scales.

---

> ### Comment · Reviewer_13ae · 2025-08-06
>
> Thank you for the further response. Regarding the novelty and motivation, using multiple networks to improve the stability of learning algorithms is a well-established technique. Such methods can allow for a larger learning rate while maintaining stability, potentially leading to faster convergence.
>
> Besides, in this case, the significantly increased computational cost limits the practicality of the approach. As a result, it is unlikely to lead to substantial improvements or widespread adoption in practice.
>
> I will also take into account the comments from other reviewers, but for now, I will maintain my current rating.

---

> > ### Author Response · Authors · 2025-08-06
> >
> > We appreciate reviewer's prompt response that gives us the opportunity of clarifying this study. As stated in the literature review, many recent studies such as TD3, TOP-TD3, DARC, REDQ, and MA-TD3 aimed at ameliorating the issue of estimation accuracy through ensemble critics for improving the quality. We think the development of ensemble critics is an ongoing issue, and thus this study introduces a novel architecture with quantile-based ensemble strategy also for addressing this issue.
> >
> > Regarding the concerns on scalability and computational cost, we still want to emphasize that the cost at interaction to environment is the key to the overall running time. The following table shows the overall running time under different interaction frequency. The interaction frequency influences the running time of each environmental step. It is apparent that as the interaction frequency decreases (cost at interaction increases), the overall running time for all methods increases. The overhead of the MAMC in comparison to baselines may becomes insignificant as the cost at interaction increases. Also, to the best of our knowledge, along the evolution of RL methods based on ensemble critics, a clear trend is the improving estimation accuracy usually comes from more critics and/or more computational cost with different integration stategies. Therefore, we believe the quality and applicability of the MAMC is guaranteed in practice.
> >
> > |Env.       |Interaction Frequency (Hz)|TD3-SMR|DARC-SMR|SAC-SMR|REDQ-SMR|dMAMC |sMAMC |
> > |----------:|-------------------------:|------:|-------:|------:|-------:|-----:|-----:|
> > |Humanoid-v5|5000                      |16.86  |40.38   |32.63  |78.76   |210.26|294.54|
> > |Humanoid-v5|100                       |26.66  |50.18   |42.43  |88.56   |220.06|304.34|
> > |Humanoid-v5|10                        |116.66 |140.18  |132.43 |178.56  |310.06|394.34|
> > |Humanoid-v5|1                         |1016.66|1040.18 |1032.43|1078.56 |1210.06|1294.34|

---

### Decision · Program_Chairs · 2025-09-17

**Decision:**

Reject

**Comment:**

The submission presents MAMC, a multi-actor multi-critic reinforcement learning framework that introduces a novel actor selection strategy based on balancing “skill” and “creativity.” Experimental evaluation on several continuous control benchmarks shows competitive performance, and an ablation study highlights the benefit of the selection mechanism. While these results show potential, reviewers found that the current technical contribution focused mainly on integrating and optimizing existing actor–critic components rather than offering a fundamentally new theoretical paradigm.

A major limitation lies in the evaluation scope. Only five scenarios are considered, which may not be insufficient to verify the method’s generality. Key related methods in the multi-actor multi-critic family are not included in the comparisons. It is also unclear whether the reported gains stem from the proposed algorithmic improvements or from increased model capacity. Additionally, the exploration–exploitation trade-off appears to lack deeper theoretical analysis. Meanwhile, concerns about performance degradation in longer training phases and weaker results compared to some competing methods remain unaddressed.

During the discussion and rebuttal, reviewers raised concerns about missing baselines, narrow evaluation scope, parameter sensitivity, unclear sources of performance gains, and insufficient analysis of the exploration–exploitation dynamics. They also pointed out cases where MAMC underperformed in later training stages and questioned the deterministic framework’s limitations. The authors’ rebuttal provided qualitative explanations but no new experiments or quantitative evidence, leaving most concerns unresolved (or only partially addressed). Furthermore, reviewers questioned the clear rationale for why the mid quantile was chosen over the mean in Eq. (6). However, the authors did not seem to offer sufficient empirical or theoretical evidence showing its specific advantages (e.g., robustness to outliers, mitigation of overestimation bias).

Given the incremental nature of the contribution, the limited and selective empirical coverage, and the lack of substantive new evidence addressing major reviewer concerns, the paper does not seem to be ready for publication in its current form. Broader evaluation, stronger comparative analysis, and deeper insight into the claimed advantages may be necessary to strengthen its overall quality.